# A conserved NR5A1-responsive enhancer regulates *SRY* in testis-determination

**Denis Houzelstein** [1,2] ✉, **Caroline Eozenou** [1,2,15], **Carlos F. Lagos** [3,4], **Maëva Elzaiat** [1,2], **Joelle Bignon-Topalovic** [1,2], **Inma Gonzalez** [2,5], **Vincent Laville** [2,6,7], **Laurène Schlick** [1,2], **Somboon Wankanit** [1,2,8], **Prochi Madon** [9], **Jyotsna Kirtane** [10], **Arundhati Athalye** [9], **Federica Buonocore** [11], **Stéphanie Bigou** [12], **Gerard S. Conway** [13], **Delphine Bohl** [12,14], **John C. Achermann** [11], **Anu Bashamboo** [1,2] & **Ken McElreavey** [1,2] ✉

The Y-linked *SRY* gene initiates mammalian testis-determination. However, how the expression of *SRY* is regulated remains elusive. Here, we demonstrate that a conserved steroidogenic factor-1 (SF-1)/NR5A1 binding enhancer is required for appropriate *SRY* expression to initiate testis-determination in humans. Comparative sequence analysis of *SRY* 5′ regions in mammals identified an evolutionary conserved SF-1/NR5A1-binding motif within a 250 bp region of open chromatin located 5 kilobases upstream of the *SRY* transcription start site. Genomic analysis of 46,XY individuals with disrupted testis-determination, including a large multigenerational family, identified unique single-base substitutions of highly conserved residues within the SF-1/NR5A1-binding element. In silico modelling and in vitro assays demonstrate the enhancer properties of the NR5A1 motif. Deletion of this hemizygous element by genome-editing, in a novel in vitro cellular model recapitulating human Sertoli cell formation, resulted in a significant reduction in expression of *SRY*. Therefore, human NR5A1 acts as a regulatory switch between testis and ovary development by upregulating *SRY* expression, a role that may predate the eutherian radiation. We show that disruption of an enhancer can phenocopy variants in the coding regions of *SRY* that cause human testis dysgenesis. Since disease causing variants in enhancers are currently rare, the regulation of gene expression in testis-determination offers a paradigm to define enhancer activity in a key developmental process.

In mammals, heteromorphic sex chromosomes (XY) evolved between 166 and 148 million years ago (MYA) and have been conserved in most species[1–3]. An XY karyotype typically leads to testis development and a male phenotype, whereas an XX complement is typically found in females. In the human 46,XY embryo, the developing gonad remains bipotential until 40 days post-conception (dpc) when the expression of the key Y-linked testis-determining gene, *SRY*, is induced in a subset of somatic cells of the gonadal primordium[4]. *SRY* expression peaks at about 44 dpc and is downregulated from 53–55 dpc to reach the baseline transcript level by 61 dpc[5–7]. *SRY/Sry* encodes a transcription factor that acts as a switch to tip the balance of antagonistic pro-testis and pro-ovarian gene networks towards a testis fate. This initial bias is amplified by a positive feedback loop between SOX9 and FGF9, which results in repression of the pro-ovarian *WNT4* signaling, leading to

---

Sertoli cell formation[4]. Single-cell sequencing analysis of human developing gonadal cells indicates that Sertoli and interstitial cells originate from a common heterogeneous progenitor pool, which then resolves into fetal Sertoli cells or interstitial cells that include Leydig cells. The data suggest that in humans Leydig and Sertoli cell specification occurs at or near the same developmental time[4]. Pathogenic variants in genes involved in testis formation, including *SRY*, cause 46,XY differences/disorders of sex development (DSD) in humans[8–10]. The spectrum of phenotypes associated with DSD in 46,XY individuals can range from a phenotypic girl with complete testicular dysgenesis to atypical genitalia/hypospadias in a boy.

Although *SRY* was identified more than 30 years ago, relatively little is known about the regulation of its expression during testis-determination. The identification of 46,XX phenotypic males with testes, due to a translocation of a 35 kilobases (kb) Y-chromosome fragment containing *SRY*, showed that this locus is sufficient for testis development in humans[11]. Subsequent efforts to understand the regulation of *SRY* focused on 2 kb of proximal promoter sequences, but the identity and role of potential enhancer element(s) remains unknown[12]. Three transcription factors, GATA4 (GATA-binding factor 4), WT1 (Wilms Tumor 1), and NR5A1 (nuclear receptor subfamily 5 group A Member 1, also known as steroidogenic factor 1, SF1, SF-1), have been postulated to play a central role in *SRY* regulation but formal evidence is still lacking[2,12,13]. Although pathogenic variants of the NR5A1 protein, a key nuclear receptor encoded by the *NR5A1* gene, are currently the most common molecular cause of gonadal dysgenesis in human[14], whether NR5A1 directly participates in the regulation of the *SRY* gene, or downstream targets in testis-determination remains to be demonstrated.

In this study, using comparative genomic analysis, we identified a conserved NR5A1-binding enhancer element located 5′ to the *SRY* gene in mammals. Genomic analysis revealed two different hemizygous base-pair substitutions involving highly conserved residues within this NR5A1-binding site, one in a sporadic case of XY sex-reversal and the other in a large familial case of Y-linked 46,XY DSD. In silico modeling and in vitro assays of these variants supported the enhancer properties of the NR5A1-responsive element. Deletion of this element by genome-editing in a novel in vitro cellular model recapitulating human Sertoli cell formation results in a significant reduction in expression of *SRY*. Taken together, our genomic data, combined with functional analysis, establish this NR5A1 enhancer element as a regulatory switch in sex-determination.

## Results

### Comparative genomics defines a conserved NR5A1-binding element in a region of open chromatin upstream of the *SRY* gene in eutherian species

In order to identify key evolutionarily conserved elements that are required for appropriate *SRY* expression during testis-determination, we first undertook a detailed comparative sequence analysis of the *SRY* locus. In the human, *SRY* is a single exon gene derived from the duplication and divergence of an ancestral *SOX3* gene after the Theria/Monotreme phyla diverged about 150 MYA[15]. *SRY* encodes a functionally conserved high mobility group (HMG) box-containing protein with limited sequence conservation in regions flanking the DNA-binding motif. The human *SRY* gene is located on the Y chromosome reverse strand (GRCh38_p13_chrY:2,786,855-2,787,682, www.ensembl.org), 5.5 kb from the pseudo-autosomal region towards the short arm telomere, and 54 kb from *RPS4Y1*, its closest neighboring gene towards the centromere (Fig. 1). Forty-five kb of the intergenic sequences between *SRY* and *RSP4Y1* consist of densely packed repetitive sequences starting 6384 bp from the *SRY* Transcription Start Site (TSS) (RepeatMasker[16]—Fig. 1a and Supplementary Data 1). A pseudogene (*RNASEH2CP1*) is present 2122 bp 5′ to the *SRY* TSS (Fig. 1a, b and Supplementary Data 1, 2) in the primate

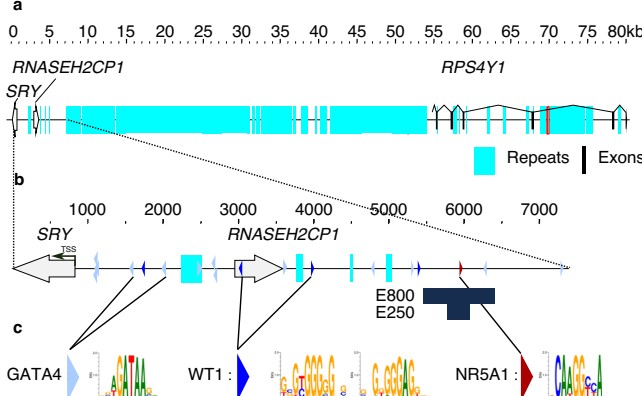

**Fig. 1 | Organization of the *SRY* locus in human. a** A sequence fragment spanning 80 kilobases (GRC38h38.p13-chrY: 2786855-2867268) from *SRY* to *RPSY4*. The intergenic region is predominantly formed of highly repetitive elements. **b** Zoomed-in view of a subregion from **a** covering the sequence from *SRY* to the beginning of the repetitive elements (GRC38h38.p13-chrY:2,786,855- 2,794,240). E800 corresponds to the region sequenced in 46,XY individuals with unexplained gonadal dysgenesis described in Fig. 3. E250 corresponds to the 250 bp fragment used for the luciferase assay described in Fig. 4E. The light blue, dark blue, and red triangles represent consensus binding sites for GATA4, WT1, and NR5A1, respectively. Turquoise indicates repetitive sequences. Open arrows represent *SRY* and *RNASEH2CP1*, while black boxes represent the *RPS4Y1* exons. The transcription Start Site (TSS) of *SRY* is indicated by an arrow. **c** Consensus binding sites for GATA4, WT1, and NR5A1 predicted by Matinspector (from the Genomatix suite).

Catarrhini clade only (Supplementary Data 3–6). It is absent, for example, from the Capuchin which belongs in the Platyrrhini, the Catarrhini sister group (Fig. 2B and Supplementary Fig. 1). Comparative sequence analysis indicates that the synteny of *SRY* and *RPS4Y1* genes as immediate neighbors is present only in the Primatomorpha, with the Greater Bamboo Lemur (Strepsirrhini, Lemuriformes, Supplementary Data 4–6) the most distant relative for which the synteny is observed. These findings of limited conservation of the synteny, associated with the fact that only 35 kb of *SRY* sequence is sufficient to drive testis-determination[11], suggested to us that long-range sequence elements required for the regulation of *SRY* expression during testis-determination are unlikely to be conserved throughout evolution. Since key regulatory elements are also unlikely to be located within the region of densely packed repeated sequences, we focused our screen for enhancer motifs on the ~7 kb of unique sequences upstream of *SRY*.

Using MatInspector (https://www.genomatix.de/), we identified 13 GATA4 and four WT1 predicted binding sites in sequences spanning 7 kb upstream of the human *SRY* TSS (Figs. 1b, c, 2a and Supplementary Data 1, 2). The eight proximal GATA4-binding sites and a single WT1-binding site, 918 bp from the TSS, have been reported elsewhere[17,18]. In contrast to WT1 and GATA4, the MatInspector analysis predicted a unique NR5A1-binding site (GRCh38_p14_ChrY:2,792,790-2,792,804) within the 7 kb region, located 5114 bp from the *SRY* TSS (Fig. 1b, c, shown in red). A subsequent analysis utilizing JASPAR, a widely employed open-access database and resource for transcription factor binding profiles, corroborated the presence of the NR5A1-binding site initially identified through MatInspector, along with two additional sites in its immediate vicinity (GRCh38.p13-chrY:2,792,742-2,792,752 and GRCh38.p14-chrY:2,792,764-2,792,774).

By using DNase-seq analysis of testis from a pool of healthy human male fetuses (ENCODE project, https://www.encodeproject.org/experiments/ENCSR729DRB/, Fig. 2a[19]), we identified two main regions of open chromatin at the *SRY* locus. The first region covers the *SRY* Open Reading Frame (ORF, GRCh38_p13_ChrY: 2,786,989–2,787,604) and the proximal flanking

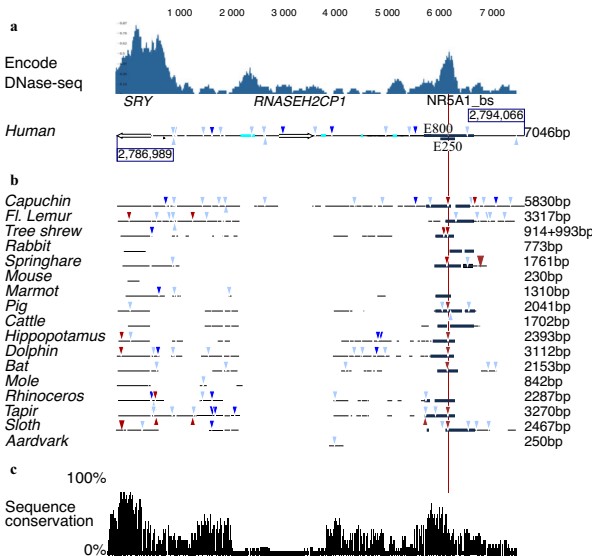

**Fig. 2 | Sequence conservation at the *SRY* locus. a** The profile of accessible chromatin, determined by DNase-seq, was downloaded from the Encode project (https://www.encodeproject.org/, experiment ENCSR729DRB, embryonic human testis). It reveals two distinct regions of accessible chromatin, one encompassing the *SRY* gene and the other corresponding to the E250 region, centered on the predicted NR5A1 binding site (NR5A1_bs). The graphical representation of the human *SRY* locus is shown below, in phase with the DNase-seq profile, and the corresponding number of base pairs is indicated on the right. GATA4 (light blue), WT1 (dark blue), and NR5A1 (red) Matinspector predicted binding sites are indicated as in Fig. 1b. E800 corresponds to the region sequenced in 46,XY individuals with unexplained gonadal dysgenesis. E250 corresponds to the 250 bp fragment used in the luciferase assay (Fig. 4e). Repetitive sequences are shown in turquoise, the *SRY* gene, and *RNASEH2CP1* pseudogene Open Reading Frame as open arrows. **b** DNA fragments from eutherian representative species, aligned to the 7046 bp human sequence shown in a. The sequences homologous to E800 are shown in black. The conservation of these sequences across a diverse range of species, including sloth, human, springhare, pig, bat, and tapir, indicates that this region was already present in their last common ancestor. Similarly, the presence of an NR5A1 predicted binding site in a homologous position shown by a red line in these species suggests a role for this NR5A1 binding site early in eutherian radiation. The total number of bases aligned to the human sequence for each species is indicated on the right. **c** The species presented in Fig. 2b were selected to represent diversity in the eutherian radiation. An estimate of sequence conservation is given. Where sequences could be aligned, a 100% conservation means that a nucleotide is conserved in all the 18 sequences. A conservation of 50% would mean that for a given aligned nucleotide, it is conserved in only half of the species. The *SRY* gene and E800 sequences both show the highest percentage of sequence conservation in the region.

sequences as expected for a gene expressed in the testis. The second region, that covers 262 bp (GRCh38_p13_ChrY: 2,792,664-2,792,925), is centered on the predicted NR5A1-binding site. Unlike in the fetal testis, where E250 is clearly accessible, accessibility was generally very modest at best in most tissues, with weak detectability demonstrated in only two out of the 733 biological samples studied (Wouter Meuleman, personal communication). The lack of accessibility of E250 in most of tissues may play a role in the specific expression of *SRY* in pre-Sertoli cells. Single-cell ATAC-seq data from somatic cells in the human XY gonad during testis-determination (6–7 post-conception weeks) confirmed the presence of open chromatin in this region in the XY supporting cell lineage GRCh38_p13_chrY-2,792,497-2,792,918; https://www.reproductivecellatlas.org/gonads/human-somatic/)[20]. Consequently, this 250 bp long DNA region is accessible in vivo at the moment of human testis-determination. This element will subsequently be referred to as E250 (Figs. 1b, 2b). Enhancers are activated by combinations of transcription factors such that the binding of only one

or a few transcription factors is possibly insufficient to activate transcription[21]. Several transcription factors with the potential to bind E250 identified by MatInspector are therefore listed in Supplementary Data 7.

To further address the evolution of this response element, we annotated *SRY* and available upstream sequences from 83 mammalian species (Supplementary Data 3–6). The human sequence is presented in Fig. 2a, and a subset of 17 representative sequences in Fig. 2b. Several attempts at aligning *Sry* 5′ sequences have been reported[12,22,23]. However, conserved regulatory elements have not been identified. We postulate that this may be due to the fact that, in every species, the region 5′ to *Sry* is formed by the reassortment of ancestral sequence elements with clade/species-specific rearrangements. This results in a complex mosaic of ancestral fragments that can be aligned because of their shared descent, together with derived fragments that cannot be aligned. To identify sequence homology to human, we performed a BLAST search with the 7046 bp of human sequence, against genomic fragments containing the *Sry* gene from the widest possible range of species (Fig. 2b and Supplementary Data 4, 5). The aligned fragments vary from one species to another both in number and size of fragments. As expected, more sequences from the closely related capuchin (5830 bp out of 7046 bp, 82%) than from the distantly related afrotherian aardvark (250 bp, 3.5%) could be aligned to humans. Such a correlation is not an absolute rule. For instance, the Glire clade, containing species such as mice and rabbits, is the sister group to the Euarchonta clade containing primates and shrews. Despite this close relationship, sampled species from Glires showed very divergent *Sry* 5′ sequences compared to both species from the Euarchonta and Laurasiatheria clades: mouse (230 bp aligned, 3.5%) or rabbit (773 bp aligned, 11%) have less and shorter homologous fragments than, for example, dolphin (3112 bp aligned, 44%) or tapir (3270 bp aligned, 46.5%). As well as sharing limited regions of homology with human 5′ *SRY* sequences, the mouse and rabbit sequences show no obvious homology to each other. Mammalian clades show a wide range of diversity, in terms of their assortment of ancestral and derived fragments that could not be predicted based on phylogenetic relationships alone. Of the laboratory models, only sequences 5′ to the pig *Sry* gene show similarity with the human (2041 bp aligned, 29%).

To explore potentially conserved responsive elements further, an eight hundred bp sequence [E800-GRCh38_p13_ChrY: 2,792,397-2,793,196] (Fig. 1b) centered on E250 was used as a query in a BLAST search to identify sequences homologous to the region of open chromatin identified by Dnase-seq in human[19]. Similar sequences were identified in a number of species from divergent clades (Fig. 2b and Supplementary Data 3–6). In particular, homologous sequences were identified in two species of sloth, members of the Xenarthra clade, that diverged from the human lineage more than 100 MYA[24] proving this DNA fragment to be at least as old as the eutherian lineage. Where the E800 homolog is present, its distance from the *Sry* TSS seems to be relatively constant, suggesting that topological constraints may exist. Notably, a predicted NR5A1-binding site, orthologous to the human NR5A1-binding site, was found in a wide array of species, including sloth (red vertical line in Fig. 2b). This result suggests that the presence of this NR5A1-binding site upstream of *Sry* predates the eutherian radiation and that NR5A1 has an ancient and conserved role as a key regulator of *Sry* expression to initiate mammalian testis-determination.

### Single nucleotide substitutions of highly conserved residues located within the NR5A1-binding element are associated with 46,XY gonadal dysgenesis

Given these findings, we hypothesized that variants within the evolutionarily conserved region in E800 would impact human testis-determination and development. To test this, we sequenced the E800

element (encompassing E250) in 358 individuals with unexplained 46,XY gonadal dysgenesis by either Sanger sequencing or whole genome sequencing. In this cohort, we identified two different and unique sequence variants in the *SRY* 5′ flanking region. Remarkably, both of these variants are located within the proposed E250 NR5A1-binding site. The first variant (Variant-1: GRCh38_p13_chrY:2,792,795, A > G) was identified in a large family of individuals with 46,XY DSD, and affected members spanning at least six generations. The clinical phenotypes of this family have been partially reported previously[25]. The family consists of two pedigrees (Fig. 3a, b), and familial recollection indicates that the two pedigrees share a common ancestor. The mode of inheritance of the phenotype is consistent with Y-linked transmission. The grouped pedigree has 106 individuals, 48 of them obligate carriers of the pathogenic variant associated with the phenotype and a penetrance of 37.5% (18/48 – Fig. 3c). The expressivity is variable with 7 affected 46,XY individuals raised as boys, and 11 as girls. All affected individuals have confirmed or suspected 46,XY complete (CGD) or partial (PGD) gonadal dysgenesis indicating disruption of testis-determination (ref. 25 and file Supplementary Table 1). Genome sequencing of three affected individuals (A-V.11, B-IV.4, B-V.6) identified the shared A > G hemizygous substitution in a highly conserved residue within the NR5A1 binding site (Fig. 4a and Supplementary Fig. 2). Sanger sequencing indicated that five other affected members of the family who were analyzed carried this variant (individuals A-VI.3, B-III.2, B-V.1, B-V.2, B-VI.1, Supplementary Fig. 2). Neither genome sequencing nor array-based comparative genomic hybridization (aCGH) revealed other genetic variants associated with DSD. In addition to this large pedigree, an independent second variant (Variant-2: GRCh38_p13_chrY:2,792,798, G > A) was identified by Sanger sequencing of the E800 element in a woman with 46,XY CGD and affects another highly conserved residue within the predicted NR5A1-binding site (Fig. 4a – Supplementary Fig. 2). The FABIAN-variant website uses transcription factor flexible models (TFFMs) and position weight matrices (PWMs) to predict the effect of DNA variants on transcription factor binding. It returns a combined score indicating to which degree transcription factor binding may be affected in a variant. (https://www.genecascade.org/fabian/)[26]. It predicts a strong detrimental effect of both Variants on NR5A1 binding (joint score of −0,8913 for Variant-1 and −0,9197 for Variant-2). The analysis of human single-cell expression data from the developing human XY gonad https://www.reproductivecellatlas.org/gonads/human-main-male/[20]) shows that *NR5A1* is co-expressed with *SRY* in human Sertoli cells (Supplementary Fig. 3) which is consistent with a role for the factor in the initiation of human testis-determination.

The Y chromosome carrying Variant-1 belongs to the H1a2a1 haplogroup, which is relatively frequent in India (Supplementary Data 8). The Y chromosome carrying Variant-2 is characterized by the Y-13355944-A-G variant that defines haplogroup E-M215 (E1b1b), which is present at high frequencies in Near East and North African populations (https://gnomad.broadinstitute.org/variant/Y-13355944-A-G?dataset=gnomad_r3,[27]). The two substitutions in the predicted NR5A1-binding motif are absent from public databases of human variants (gnomAD v3.1.2, gnomad.broadinstitute.org, July 2023, 76,156 genomes from unrelated individuals sequenced as part of various disease-specific and population genetic studies). The two potentially pathogenic variants in E250 are present in two well-characterized, independent Y chromosome haplogroups and thus are predicted to affect testis-determination in distinct genetic contexts.

### In silico modeling predicts that single nucleotide substitutions disrupt the interaction between the NR5A1 protein and the response element

In silico computational analysis was performed to assess the effect of these two variants on interactions between NR5A1 and its E250

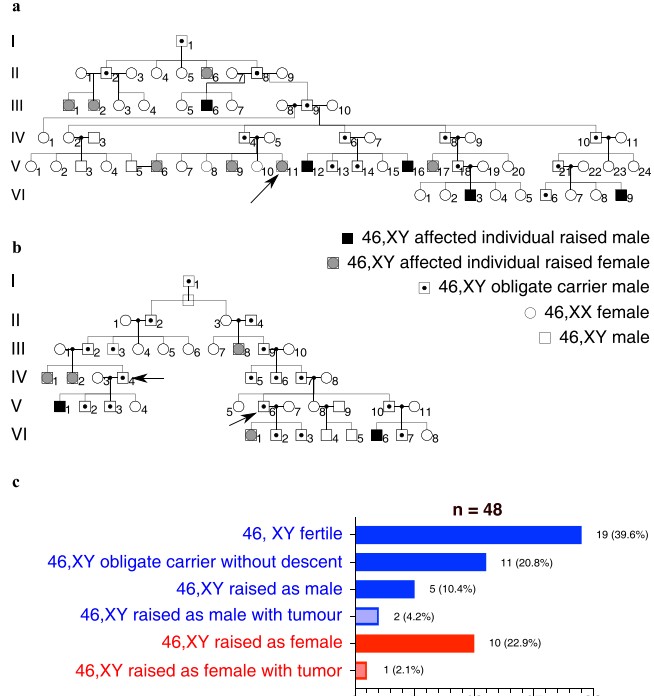

**Fig. 3 | Pedigrees of a familial case of 46,XY disorder of sex development (DSD). a, b** Two pedigrees illustrating the two branches of a family from the same community, presenting with 46,XY gonadal dysgenesis. Arrows indicate the three individuals for whom the whole genome sequence was obtained. **c** Distribution of phenotypes, incidence of gonadal tumors, and assigned sex of individuals with the SRY variant.

response element. NR5A1 is unusual as it belongs to a small subgroup of nuclear receptors that preferentially bind to DNA response elements as a monomer, rather than as a homo- or hetero-dimer[28]. NR5A1 binds to variations on a CCA AGGC(or T)C(or T)A motif (designated −3 to +6) (Fig. 4a). To increase kinetic stability, monomeric nuclear receptors typically have a supportive "A"-box region that interacts with the minor groove of DNA (CCA; −3 to −1) in addition to the core interface between the "P"-box and the major groove motif (+1 to +6) (Fig. 4a, b). Based on the crystal structure of NR5A1 bound with mouse inhibin-α target sequence, the residue altered in Variant-1 (A > G, −1, purine to purine) is predicted to interact with the "A"-box motif of NR5A1 (shown by $G_{-1}$ and corresponding $C_{-1}$ residue in cyan, Fig. 4c). In contrast Variant-2 (G > C, +3, purine to pyrimidine) directly disrupts interactions with the "P"-box of NR5A1 (shown by $C_{+3}$ and corresponding $G_{+3}$ residue in magenta, Fig. 4d) and induces a conformational change that displaces the "P"-box alpha helix. The disruptive effects of these two variants were supported by simulation studies showing less flexibility of bound NR5A1-DNA compared to wild-type, and a reduced number of hydrogen bonds between variant DNA response elements and the NR5A1 protein (Supplementary Fig. 4).

### In vitro reporter assays show that the variants are associated with a significant reduction in reporter activity

In vitro studies of function initially focused on the ability of E250 to activate gene expression in a luciferase reporter assay system (Reference fragment corresponding to the published human sequence, GRCh38_p13_chrY:2792658-2792907, E250 in Figs. 1, 2b and Supplementary Data 9−14). In co-transfection studies with NR5A1, a significant increase in reporter activity was observed using the wild-type E250 sequence ($p = 1.379e{-}08$, Wilcoxon rank-sum exact test in HEK293T cells in Fig. 4e and Supplementary

Data 15, and in HeLa cells in Supplementary Data 16). When either Variant-1 (A > G), or Variant-2 (G > C) was introduced, or the NR5A1-binding site was deleted (ΔNR5A1: CCAAGGCT > CCTCGAGT)

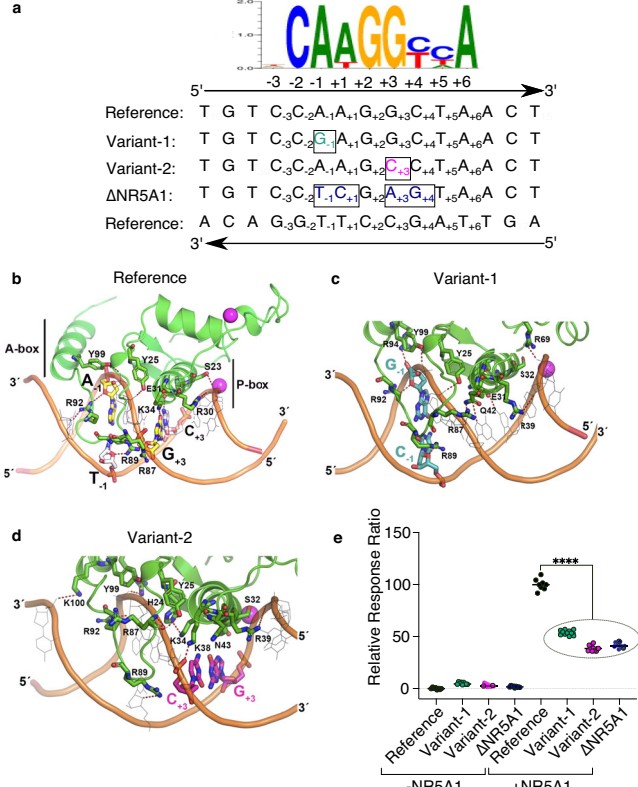

**Fig. 4 | In silico and in vitro analysis of E250 function. a** Alignment of the sequence flanking the NR5A1-binding site (GRCh38-chrY: 2,792,790-2,792,804), in the reference genome (Reference), mutated in the familial case (Variant-1), in the sporadic case (Variant-2), and in a negative control with four substitutions disrupting the core sequence (ΔNR5A1). The NR5A1 consensus binding site is shown above for comparison, featuring the core element from +1 to +6 and the flanking sequence from −1 to −3. **b**–**d** Schematic representation of NR5A1 DNA-binding domain (DBD) interactions with E250 variants based on the crystal structure of NR5A1 bound to the inhibin-A promoter. Protein residues of NR5A1 are depicted in green, while the DNA backbone is shown in orange. Hydrogen bond contacts are represented as red dashed lines. **b** Structural model of wild-type NR5A1 bound to DNA. The two paired binding site residues of interest are shown, based on the sequences in panel A (A-1/T- 1 and G + 3/C + 3). NR5A1 binds to DNA primarily as a monomer. The protein P-box region (codons 31 to 35) interacts directly with the core binding motif (+1 to +6), whereas the A-box protein region (codons 89 to 92) interacts with the flanking DNA sequence (−1 to −3). **c, d** Structural models of wild-type NR5A1 bound to variant response elements associated with testicular dysgenesis. In Variant-1 (panel **c**), the A > G purine to purine change at nucleotide −1 in the flanking sequence (shown in cyan, with corresponding reverse strand T > C change) affects interactions between the DNA and A-box residues changing the H-bond pattern between DNA and basic residues R87, R89, R92, R94, and Y99 of NR5A1. In contrast, with Variant-2 (panel **d**), the G > C purine to pyrimidine change at nucleotide +3 (shown in magenta, with corresponding reverse strand C > G change) affects interactions between the DNA and P-box. There is disruption of critical H-bond interactions that would typically occur between NR5A1 codon K38 and the DNA amine group of G9, and between codon E31 with the amine group of C22; furthermore, the amine group of codon K34 is neutralized by H24 and E31 in NR5A1. These changes induce a displacement of both "A-box" and "P-box" helices that may influence binding affinity or kinetics. **e** Transient gene transfection assay showing activation of a luciferase reporter construct containing the wild-type or variant E250 response element co-transfected with or without NR5A1 in HEK293T cells (12 technical replicates, outliers identified by the interquartile Range (IQR) method and then removed, with comparison performed using the Wilcoxon rank-sum exact test (p value = 1.379e-08****) in R.

(Fig. 4a), reporter activity was significantly reduced (p < 0.0001, Fig. 4e). Plasmid sequences, maps, luciferase assay results, and statistical analysis are provided in Supplementary Data 9–16 and Source Data.

The E250 NR5A1-responsive enhancer element is not conserved in mice, therefore, to assess the role of the E250 element in a biologically relevant context, we used a recently developed protocol that differentiates human-induced Pluripotent Stem Cells (hiPSCs) towards gonadal progenitors by using sequential changes in defined culture medium (M1, M2, and M3 in Fig. 5). A comprehensive description of the characteristics of this in vitro model has been published elsewhere[29]. It circumvents the need for forced continuous expression of exogenous factors for Sertoli-like cell formation and has already been used to model a genetic variant causing 46,XY gonadal dysgenesis. We confirmed that *SRY* was expressed in hiPSCs as previously described in ref. 30 (day 0 of differentiation, Supplementary Fig. 5), and we show that the expression of *SRY* in 46,XY wild-type cells was at a basal level after 36 h of differentiation (Medium1 – 36 h in M1, Fig. 5). The change from M1 to M2 medium saw a rapid induction of *SRY* expression (<6 h, Fig. 5a). *SRY* expression peaked at 3.0 days of differentiation (36 h in M2) and then returned to a near basal level by 5.5 days of differentiation (48 h in M3, Fig. 5a). In wild-type 46,XY cells, *SOX9* was expressed at a basal level both in hiPSCs and after 1.5 days of differentiation (Fig. 5a). During the differentiation process, *SOX9* expression increased and was delayed compared with *SRY*, consistent with *SOX9* being a direct downstream target of SRY (Fig. 5a). In this hiPSC differentiation model the expression of *SRY* and its target *SOX9* can therefore be induced within 12 h of a medium change. In wild-type cells *WNT4* expression initially increased, as expected for a gene contributing to the proliferation of gonadal progenitors in both sexes, and prior to becoming an ovarian marker[31] (Fig. 5a). The levels of *NR5A1* (Supplementary Fig. 6), the testis marker *FGF9* (Supplementary Fig. 7), and the pro-ovarian gene *FOXL2* (Supplementary Fig. 8) stayed stable during the period of *SRY* increase. The Sertoli cell marker *AMH* was undetectable (Supplementary Data 17); however, we previously demonstrated a significant increase in its expression at later stages of differentiation[29].

From our wild-type reference hiPSC cell line, we generated a hiPSC clone carrying a 33 bp deletion encompassing the NR5A1-binding site (E250-Δ33, GRCh38_p13_chrY: 2,792,792-2,792,824). At the initial analysis point (M1-36 h), mutant cells carrying a deletion of the NR5A1-binding motif showed lower relative expression of *SRY* (reduction by 40 to 70%, Supplementary Fig. 9) compared to wild-type control cells (Fig. 5b, boxed area, p < 1e-04 at M1-36 and p = 3.05e-02 at M2-06, Linear mixed model fit by REML. t-tests use Satterthwaite's method [## lmerModLmerTest]; Supplementary Data 18–23). This attenuation in relative *SRY* expression persisted during the critical early M2 phase of *SRY* induction (Fig. 5a, b), although a delayed increase in relative *SRY* expression was observed in mutant cells subsequently (Fig. 5b) Taken together, these data indicate that, in vitro, the activation of E250 by NR5A1 is essential for the induction of *SRY* expression, but compensatory mechanisms occur at later stages. Even small changes in SRY dosage at critical timepoints can have marked effects, as highlighted by the description of a family of five closely related members with an *SRY* variant, in which a twofold reduction in SRY protein activity results in 46,XY DSD with incomplete penetrance[32]. During the differentiation process, neither *SOX9*, *WNT4*, *NR5A1*, nor *FOXL2* expression exhibited any significant change between the wild-type and mutant cells at any of the timepoints (Supplementary Fig. 10). Deletion of the NR5A1-binding site in E250 therefore resulted in a significant reduction of *SRY* expression at a critical timepoint in an in vitro model of human Sertoli-like cell differentiation, which together with our other data, indicate that E250 acts as an enhancer of *SRY* during human Sertoli cell differentiation.

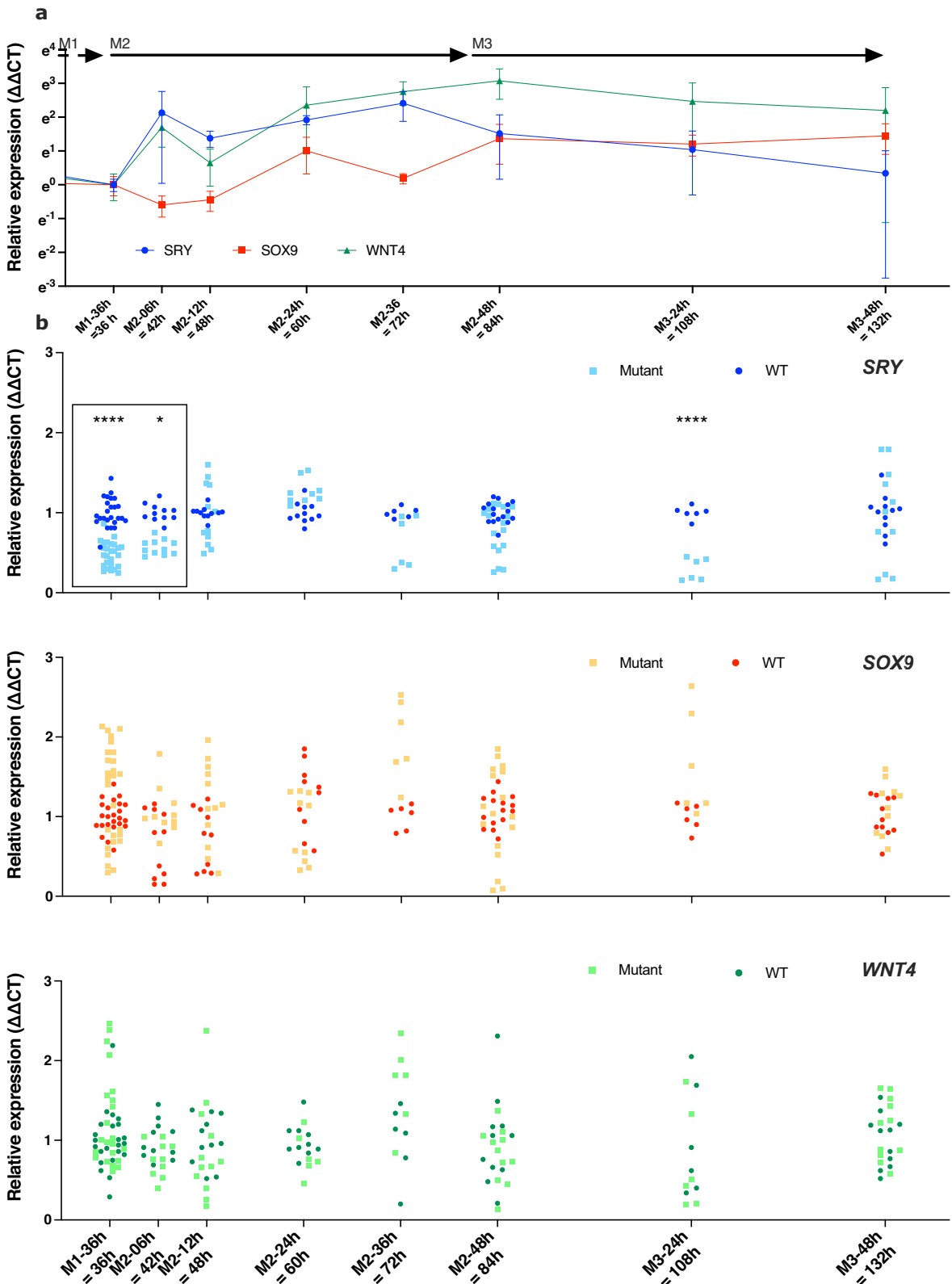

## Discussion

Using comparative sequence analysis and genomic sequencing of 46,XY individuals with disrupted testis-determination, we have identified the first known enhancer (E250) of the mammalian testis-determining gene *SRY*. Located within a region of open chromatin, ~5 kb upstream of *SRY*, E250 contains an evolutionary conserved NR5A1-binding site. A combination of in silico modeling, in vitro assays

and a novel in vitro model of human Sertoli cells, indicates that the NR5A1-binding site is required for appropriate *SRY* expression during human testis-determination. We have recently developed an in vitro model of sequential differentiation of human-induced pluripotent stem cells (hiPSCs) towards Sertoli-like cells, using only a defined culture medium that was successfully used to model aspects of

**Fig. 5 | Sertoli-like cell differentiation from hiPSCs. a** Differentiation kinetics of human hiPSCs derived from a healthy 46,XY male, starting with hiPSC, and sequential alteration of the culture medium (M1, M2, and M3) to induce differentiation into Sertoli-like cells. For greater clarity, the x-axis starts after 30 h in M1. The labels M1-36h, M2-06h, M2-12h, M2-24h, M2-48h, and M3-48h indicate the time cells spent in specific media, while the labels 36, 42, 48, 60, 72, 84, 108, and 132 h indicate time elapsed since the initiation of differentiation, triggered by the replacement of mTeSR with M1 medium. The expression levels of key genes, *SRY* (blue), *SOX9* (red), and *WNT4* (green) were quantified using qRT-PCR with the ΔΔCT method. Normalization was performed using the 18S rRNA *RPL19* gene, and the end of M1 served as the calibration point. **b** Comparison of *SRY*, *SOX9*, and *WNT4* expression between wild-type and mutant E250-Δ33 hiPSC cell lines. The expression levels were quantified using qRT-PCR with the ΔΔCT method using the wild-type condition as the calibrator for the visual representation. Gene expression is visually represented with wild-type clones shown in dark and mutant clones light blue for *SRY*, in red and light brown for *SOX9*, and in dark and light green for *WNT4*. The data were pooled with two to five biological replicates performed, with each experiment having three to six technical replicates. The statistical analysis was performed using a linear mixed model. At the initial timepoints (M1-36h, M2-06h), data were shown within a box, to highlight a 40 to 70% reduction in relative expression of *SRY* in mutant cells compared to wild-type control cells (detailed in the Supplementary Information file).

gonadal development with a naturally occurring human sex-reversing gene variant[29]. Using this model, we established detailed expression profiles of the main pro-testis and pro-ovarian genes during testis-determination. We show that *NR5A1* has an expression profile consistent with a role in the initiation of testis-determination that is confirmed by scRNAseq datasets. hiPSCs clones carrying a 33 bp deletion, encompassing the NR5A1-binding site, show a significant reduction in the initial expression of *SRY*, which, combined with our other data, indicates the importance of this enhancer motif at a critical moment in testis-determination.

DSDs are collectively among the most common congenital conditions, but the genetic etiology is known in less than 50% of the 46,XY DSDs that disrupt testis development[9,33]. Considering that both the timing and the threshold levels of gene expression are key for testis formation, our results suggest that pathogenic variants in regulatory elements of disease-causing genes may be an important contributor to the etiology of 46,XY DSD. Our results define a new cause of *SRY*-linked 46,XY DSD, where single-base-pair substitutions in an enhancer element of *SRY* result in a spectrum of 46,XY DSD phenotypes.

In 46,XY individuals, pathogenic variants in the coding sequence of *NR5A1* cause a wide range of reproductive pathologies ranging from gonadal dysgenesis to phenotypic males with spermatogenic failure (https://www.omim.org/entry/184757). Our results show that the 46,XY gonadal dysgenesis seen in association with *NR5A1* variants is likely to be due, at least in part, to the inability of NR5A1 variants to initiate *SRY* expression in the bipotential genital ridge. Our data support the role of NR5A1 as a key player in cell fate decision in sex-determination through the direct regulation of *SRY* expression, a role it is likely to have played at least since the last eutherian common ancestor. Indeed, the role of NR5A1 in sex-determination may be general, since it was recently recognized as being the sex-determining factor in some reptile species[34]. Our data expand the range of biological functions already attributed to NR5A1, including regulation of key steroidogenic gene expression, cell proliferation, and survival[35]. Our observations also reinforce the hypothesis that the initiation of testis-determination occurs via an *SRY* gene regulatory pathway that is sensitive to threshold levels of these factors. The inability to reach and maintain these critical thresholds leads to the repression of testicular development by pro-ovarian factors such as WNT4[2,3]. Indeed, the exquisite sensitivity to time- and dose-dependent thresholds may explain the phenotypic variability seen in the familial case reported in this study. Variable phenotypic expression in not unusual in 46,XY DSD since paternal transmission of *SRY* coding variants has been reported in at least ten families[32,36].

Enhancers play a crucial role in gene regulation, however, human developmental disorders attributed to disrupted enhancer elements are rare[37,38]. Mammalian genes involved in development are located near more enhancer elements than genes in general[39]. These multiple enhancers, termed shadow enhancers, often display overlapping or partially overlapping spatiotemporal activity on their target genes. This redundancy results in robustness during development and protects individual enhancer function against disruptive genetic variants[40]. However, testis-determination seems to be an exception, where single base-pair changes in E250 involving the NR5A1-binding site phenocopy deleterious mutations within the *SRY* coding sequence. *SRY* expression therefore shows a surprising lack of robustness when considering its essential function in determining sex[32,41]. A similar phenotypic effect has been proposed for the *SOX9* enhancers that positively regulate *SOX9* expression in testis-determination[42].

It is unlikely that NR5A1 acts alone to control *SRY* expression on the E250 enhancer. Other transcription factors may well act in concert with NR5A1 to promote *SRY* expression during testis-determination. The NR5A1-binding site is immediately flanked by two predicted consensus SOX-binding sites suggesting the possibility of a positive autoregulatory feedback loop enhancing *SRY* expression (Supplementary Data 7). It is still unclear if E250 is the only enhancer element required for appropriate *SRY* expression during testis-determination. A combination of in silico comparative genomics as well as genomic screening of the region in 46,XY DSD individuals has not identified any other putative enhancer elements, although disruptive changes have been reported in the minimal promoter region of *SRY* in association with 46,XY DSD[43–45]. Understanding spatiotemporal and quantitative expression dynamics in regulatory gene networks during embryonic development is challenging, since they are characterized by complex multi-enhancer systems. In contrast, the regulation of gene expression in testis-determination offers a simple paradigm to define and understand the mechanics of enhancer action in a key developmental process.

## Methods

### Ethical Approval and informed consent
Ethical committee approval for exome and genome studies was obtained from the Comité de Protection des Personnes, Ile-de-France (N°IRB00003835) and the National Human Genome Research Institute Institutional Review Board (15-HG-0130). Local institutional review boards also agreed to the study and written informed consent was obtained from subjects and family members.

### In silico analysis
Sequences were downloaded from NCBI. The complete human genomic region and annotations were obtained from the NC_000024 human contig. The sequences were then annotated with CLC main workbench (https://digitalinsights.qiagen.com). Sequences and their annotations are given in Supplementary Data 1–5, and can be visualized for free with the demo version of CLC. The predicted binding sites for GATA4, NR5A1, and WT1 were identified with the MatInspector module from the Genomatix suite (Precigen Bioinformatics Germany GmbH, https://www.genomatix.de/solutions/genomatix-software-suite.html). Please consult Supplementary Data 6 for the comprehensive list and localization of the predicted transcription factor binding sites identified in E250. The repeated sequences in the Y chromosome locus between *SRY* and *RPS4Y1* were identified with Repeatmasker (https://www.repeatmasker.org/cgi-bin/WEBRepeatMasker)[16].

The sequences for the *SRY* gene and its upstream sequences from 83 eutherian species were downloaded from NCBI. Their accession

number, common name and latin name are given in Supplementary Data 6. The phylogenetic tree of the Eutherian mammalian species was reconstructed from several sources[46–49] as well as from https://www.ncbi.nlm.nih.gov/taxonomy. The complete tree is shown in Supplementary Fig. 1. The annotated sequences are given as a CLC main workbench gene list in Supplementary Data 3. They were used to build Fig. 2, in which sequences from the 17 most representative species were conserved. The 17 annotated sequences are shown in Supplementary Data 4, 5.

The DNase-seq profile shown in Fig. 2a covers the GRCh38-chrY:2,786,989–2,794,066 genomic interval and was downloaded from Encode (https://www.encodeproject.org/, experiment ENCSR729DRB[19], embryonic human testis from a pool of healthy male fetuses. The precise developmental stage was unfortunately unrecorded). The sequences used to build Fig. 2b were selected on two criteria: they were in close vicinity to *SRY* and they returned a hit in a blastn search against the human *SRY* with default settings (Number of threads:8, mask low complexity regions, Expect: 10, Word size: 11, Match 2, Mismatch −3, gap costs: existence 5, extension 2). This association was considered to be a strong indicator of homology. The homologous DNA fragments obtained by this method were subsequently aligned on the 7046 bp human sequence shown in Figs. 1b, 2b using the ClustalW algorithm from the CLC main workbench and subsequently hand-edited (Supplementary Data 4, 5). These species were selected to represent as much of the eutherian diversity and dispersed as evenly as possible along the tree in order to make the sequence conservation spectrum in Fig. 2d relevant.

### Whole genome sequencing

Whole genome sequencing was carried out on the Illumina HiSeq. 2000 platform (Illumina Inc. San Diego, USA) using paired-end chemistry with a 75 base-pair read length (NovoGene, Hong Kong). Genomic DNA was fragmented by sonication to the size of 350 bp. The end-polished DNA fragments were A-tailed and ligated with the full-length adapters. The PCR products for the libraries were purified with an AMPure XP system (Beckman Coulter Life Sciences). The Genome Analysis Toolkit (GATK, version 3.7) best practices pipeline was followed to process the sequence reads from FASTQ files. The Burrows–Wheeler Aligner (BWA) was utilized to map the paired-end clean reads to the human reference genome. Picard version 1.62 129 (http://broadinstitute.github.io/picard/) and SAMtools version 0.1.18 were used to mark duplicate reads and to process the BAM file manipulations, respectively. GRCh37/hg19 was used as the human reference genome. Quality filtering, base quality score recalibration, and variant quality score recalibration were accessed in the various preprocessing tools of the GATK best practices pipeline. SNPs and indel variants were annotated to dbSNP 138 using GATK Unified Genotyper. Annotation of variants was performed with the tool ANNOVAR, including protein-coding changes, genomic regions affected by the variants, allele frequency, and deleteriousness prediction. Structural variation (SV) was called using DELLY and CNVs with control-FREEC. The Interactive Genome Viewer (IGV) (http://software.broadinstitute.org/software/igv/) was used to visually inspect the called variants.

### Case histories

A preliminary report describing 21 members of the extended family this work is based on has already been published[25]. All affected 46,XY individuals presented with 46,XY partial or complete gonadal dysgenesis. Details of the cases are provided in Supplementary Table 1.

### Sanger and whole genome sequencing

An 885 bp region covering the E800 fragment is shown in Figs. 1b, 2b was PCR amplified and Sanger sequenced from 358 individuals with 46,XY DSD of unknown etiology (GRCh38.p13-chrY: 2792451-2793335 from primers SRY-Enh-7f: TGCCTGTAAAAACAAGGTTTCATACTTGGG

and SRY-Enh-7r: AGCCCGGAATTCCTCAGATTTTTCC). Part of the amplified fragment was subsequently sequenced using an internal primer (SRY-amp4-1r: AAGAAGGCAAAAACGGGCAC).

The whole genome from three members of the family carrying Variant-1 (one affected male and two male obligate carriers) was sequenced. DNA from six more 46,XY Variant-1 obligate carriers was analyzed by Sanger sequencing. Sanger sequencing was also used to identify the variant in the sporadic Variant-2 case (Supplementary Fig. 2). She had other potential known genetic causes of DSD excluded using a targeted panel-based approach[8].

### Modeling and simulations

The solution structure of the human NR5A1 (SF-1) DNA-binding domain in complex with mouse inhibin-α target sequence (PDB id 2FF0)[28] was used as the starting point for a generation of NR5A1 [residues 10–111] and *SRY* enhancer sequence models for reference (wild-type) and clinical variants. For molecular dynamics (MD) simulations, NR5A1-*SRY* systems were solvated, and ionized using the CHARMM-GUI web server interface[50]. The protein–DNA systems were embedded in a water box of $80 \times 80 \times 80$ Å, and the system charges equilibrated with Na$^+$ and Cl$^-$ atoms up to a concentration of 0.15 M NaCl. The final systems contain -55,000 atoms. All MD simulations were performed using Gromacs v2020.5[51]. The simulation systems were first minimized for 5000 steps, followed by gradual heating from 0 to 310 K by running short MD simulations of 500 steps each cycle using the NVT ensemble. The simulation was switched to NPT conditions and was further equilibrated for 2.5 ns while constraining the protein backbone with an initial force constant of 10 kcal/(mol·Å$^2$) and gradually decreasing to 8, 6, 4, 2, 1, 0.5, 0.05 kcal/(mol·Å$^2$) every 250 ps of MD simulation. Finally, a 50 ns production MD simulation was performed at a temperature of 310 °K (V-rescale thermostat), pressure at the atmospheric NPTensemble (Parrinello-Rahman barostat), and periodic boundary conditions using the Verlet cut-off scheme. The LINCS algorithm limited all of the bond lengths[52]. The electrostatic interactions were calculated using the Particle Mesh Ewald (PME) summation scheme[53]. Trajectory analysis was performed using Visual Molecular Dynamics v1.93 (VMD) software[54]. Protein and nucleic acid conformational changes were explored by measuring the root-mean-square deviation of Cα and C5′ atoms (RMSD) and root-mean square-fluctuation of individual residues (RMSF) in both the protein and nucleic acids. We also measured the solvent-accessible surface area (SASA), and the hydrogen bond interactions established between NR5A1 and *SRY* enhancer variants through simulation. The resulting parameters for variants and reference controls were compared using a one-way analysis of variance (ANOVA) and Tukey's multiple-comparison posttest. Differences between groups were considered to be significant at a *P* value of <0.05. Statistical analyses were performed with GraphPad Prism Version 9.0 (GraphPad Software, Inc., San Diego, CA, see also Supplementary Fig. 4).

### Constructs, cell lines, transfections, and luciferase assays

The pCMX and pCMX-*NR5A1* plasmids have already been described in ref. 55. The pCMX-*NR5A1* plasmid contains a DNA fragment corresponding to bases 38 to 1910 of the NM_004959 sequence cloned into pCMX (https://www.addgene.org/vector-database/2249/). Please refer to Supplementary Data 9–14 for the sequence of the various plasmids used in this study.

For the luciferase assay, the Reference construct contains a 250 bp DNA fragment (GRCh38.p13-chrY:2792658-2792907) centered on the NR5A1-binding site and cloned into pGL4.26 (Promega #E8441, Supplementary Data 11). The Variant-1 construct contains the same sequence as the reference plasmid except for an A > G substitution at position homologous to GRCh38.p13-chrY:2,792,795, (Fig. 4a and Supplementary Data 12). The Variant-2 construct contains the same sequence as the reference plasmid except for a G > A substitution at position GRCh38.p13-chrY:2,792,798 (Fig. 4a and Supplementary

Data 13). The ΔNR5A1 plasmid contains the same sequence as the reference plasmid except that the CAAGGCT sequence has been replaced with CTCGAGT at the position homologous to GRCh38.p13-chrY:2,792,794_2,792,800 (Fig. 4a and Supplementary Data 14).

HEK293T cells were a gift from Aurélie Claes (Institut Pasteur). Cells were thawed at day 0 (d0), passaged twice (d2, d4), and seeded in a 96-well plate (Eppendorf #0030730.119) at a density of 25,000 cells per well in 100 μl medium at d6. At day 7, cells were transfected using FuGENE HD transfection reagent (Promega, #E2312) according to the manufacturer's instructions with the following specifics: Cells were grown in DMEM + glutamax (GIBCO #31966-021) supplemented with fetal bovine serum 10% final (Gibco #10270-106) and 95 ng of total DNA was transfected in each well (60 ng reporter, 15 ng of pCMX-*NR5A1* or pCMX, 15 ng pCMX, and 5 ng pRenilla (Supplementary Data 10) vector with a 3:1 ratio of FuGENE (μl): DNA (ng), and a complexing time of 15 min. About 100 μl of fresh medium was added at d8. The Luciferase and Renilla activities were measured at d9 in a Glomax multi+ detection system (Promega), on Dual-Glo settings, and the Dual-Glo® Luciferase Assay System (Promega # E2940) according to the manufacturer's instructions. HeLa cells were a gift from Jacob Seeler (Institut Pasteur). These cells were treated similarly to the HEK293T cells with the exception that the ratio of FuGENE (μl): DNA (ng) was 4:1 and that cells were seeded in a 96-well plate (Eppendorf #0030730.119) at a density of 32,000 cells per well. Luciferase raw results are given in Source Data.

Data were analyzed with R version 4.3.0 and RStudio version 2023.06.0 + 421[56,55]. Outliers were identified by the InterQuartile Range (IQR) method and then removed. Comparison between reference samples with NR5A1 and all other variants with NR5A1 pulled together was performed using a Wilcoxon rank-sum exact test (R version 4.3.0 and RStudio Version 2023.06.0 + 421; Detailed in Supplementary Data 15, 16). The final figure was assembled with GraphPad Prism Version 10.0.0 (131) for MacOS, GraphPad Software, San Diego, California USA, www.graphpad.com.

### Induced pluripotent stem cells

The 46,XY human-induced pluripotent stem cell (hiPSC) clone generation, culture, and CRISPR-CAS9 mutagenesis have already been described in ref. 57. To delete the 33pb fragment from the Y chromosome, we used CRISPR/CAS9 technology. The gRNA and HDR template were designed using the CRISPOR and the Benchling web tools (https://benchling.com/), respectively. $1 \times 10^6$ hiPSC cultured in the mTeSR Plus medium (#100-0276, StemCell Technologies) were dissociated with Accutase (#0920, StemCell Technologies), nucleofected (AMAXA 4D Nucleofector system, Core Unit AAF-1002B and X unit AAF-1002; Lonza, Switzerland) with both RNP complexes and the HDR template, and then plated on Laminin-521 (#77003, StemCell Technologies) in six-well plates. RNP complexes are a mix of 225 pmol of two RNA, a crRNA (#Alt-R® CRISPR-CAS9 crRNA, TGCTGGCTCAC-TAGACAAAG) and a tracrRNA-ATTO+ (#Alt-R® CRISPR-CAS9 tracrRNA) (all from Integrated DNA Technologies (IDT), IA, USA), with 120pmol of Cas9 protein (#Alt-R® S.P. Hifi CAS9 nuclease 3 NLS, IDT; #Alt-R® CAS9 electroporation enhancer, IDT). The HDR template is a single-stranded oligodesoxynucleotide (ssODN) (500 pmol Ultramer DNA oligo, CTCTCCATAAAATGAAGGTCACTTTTGATCTTTTC CAGGGTCTTCCTTCAGTTCCTTTTTGAGCCAGCAGCTGTTTGACCAA-GAACCATTTTAGGAAACAGTTTTTAAAGATACCTCATG, IDT). Twenty-four hours later, ATTO+ transfected hiPSC were FACS-sorted (MoFlo Astrios, Beckman Coulter; CYTO-ICAN platform, ICAN Institute, Paris, France) (around 2% transfection efficacy) and plated at very low density (10 cell/cm²) on Ln521 with CloneR supplement (#05888, StemCell Technologies) for clonal selection. One week later, 192 hiPSC clones were picked under a stereomicroscope and cultured on Laminin-521 in 96-well plates. When confluent, each hiPSC clone was duplicated for either cryopreservation or DNA extraction. The DNA of each clone was analyzed by PCR (forward primer TGTGGCTATCCATGCCTGAA and reverse primer AAAGCCCGGAATTCCTCAGA, IDT) to screen for genetically modified clones. Sanger sequencing was performed to confirm the CRISPR-CAS9 modification. 95% of the sequenced clones were genetically modified but only 1.8% had the 33 bp deletion. Two mutated clones were thawed, amplified and verified for chromosomal integrity (iCS-digitalTM PSC – 24 probes to check the most common genomic abnormalities before and after the modification; StemGenomics, Montpellier, France).

hiPSCs were cultured at 37 °C, in a 5% $CO_2$ incubator in T25 flasks coated with Matrigel (#354277, Corning® Matrigel® hESC-Qualified Matrix, LDEV-free) in mTeSR Plus medium (#100-0276, StemCell Technologies). When the hiPSCs reached sub-confluency, they were dissociated into aggregates using ReLeSR solution (#05872, StemCell Technologies). Five percent of the dissociated cells were then plated onto fresh Matrigel-coated T25 flasks in mTeSR Plus medium supplemented with thiazovivin (1000X, #130-106-542, Miltenyi Biotec) for 48 h, followed by culture in unsupplemented mTeSR Plus medium.

The 46,XY hiPSC differentiation protocol has already been described[29] with the following minor alteration. Before differentiation, cells were dissociated into very small aggregates in the ReLeSR solution (#05872, StemCell Technologies) and plated at a very low density. Cells were then allowed to recover for 24 h in mTeSR™ Plus medium (#100-0276, StemCell Technologies) with Thiazovivin (#130-106-542, Miltenyi Biotec) and then 30 h in mTeSR™ Plus without Thiazovivin. The hiPSCs were then subjected to serial differentiation in a conditioned medium with minor serial modifications of the medium composition as already described in ref. 29. Medium composition is as follows: CDM-PVA (basal medium) = Advanced DMEM/F12 (#12634010, Thermo Fisher Scientific)/Iscove's Modified Dulbecco's Media (IMDM, #31980030, Thermo Fisher Scientific), 50/50 supplemented with 0.1% w/v cold water-soluble polyvinyl alcohol (#P8136, Merck Millipore), 100 U/ml penicillin-streptomycin (#15140122, 10,000 U/ml, Thermo Fisher Scientific), 1X concentrated lipids (11905031, 1:100, Thermo Fisher Scientific), 1:25,000 monothioglycerol (MTG, #M6145, Merck Millipore) and 1:2000 transferrin (#1065220200, water-soluble, Merck Millipore).

M1 = CDM-PVA, bFGF (20 ng/ml; #233-FB, R&D), Ly294002 (10 μM, Pi3K inhibitor, #L9908, Merck Millipore) and BMP (10 ng/ml; #214-BP, R&D).

M2 = CDM-PVA, bFGF (5 ng/ml), BMP (20 ng/ml), and Retinoic Acid (100 nM, #R2625, Merck Millipore).

M3 = Advanced DMEM (#12634010, Thermo Fisher Scientific), 100 U/ml penicillin-streptomycin (#15140122, 10,000 U/ml, Thermo Fisher Scientific), 1:100 Insulin, Transferrin, Selenium (ITS, 100X #12097549, Thermo Fisher Scientific), and EGF (20 ng/mL; human recombinant #ab9697, Abcam).

### Quantitative reverse transcription polymerase chain reaction (qRT-PCR) for undifferentiated hiPSCs and derivatives

The protocol has already been described in ref. 29, see also https://assets.thermofisher.com/TFS-Assets/LSG/manuals/cms_042380.pdf and https://toptipbio.com/delta-delta-ct-pcr/ for methods and Calibrator/Normalizer definitions. TaqMan probes (Applied Biosystems) were used: the *18S rRNA RPL19* was used as the Normalizer (Housekeeper) gene (#Hs02338565_gH); *AMH* (Hs00174915_m1), *FGF9* (Hs00181829_m1), *FOXL2* (Hs00846401_s1), *NR5A1* (Hs00610436_m1), *SOX9* (Hs01001343_g1), *SRY* (Hs00976796_s1), and *WNT4* (Hs01573505_m1). A flowchart depicting the transformation of raw data into figures is provided as Supplementary Fig. 11. qRT-PCR raw results are given in Source Data. The $\Delta\Delta C_T$ method was used with the M1_36h00 condition as the Calibrator in Fig. 5a (Supplementary Data 42–48) and the wild-type cell line condition as the Calibrator in

Fig. 5b (Supplementary Data 36–41). At each timepoint in the differentiation protocol, between two and five biological replicates were studied, each with three to six technical replicates (Supplementary Fig. 7). Data were analyzed with R version 4.3.0 and RStudio Version 2023.06.0 + 421[56]. Outliers were identified by the InterQuartile Range (IQR) method and then removed (Supplementary Data 30–35). Since the qRT-PCR measurements are obtained in different experiments with an inter-experiment variability and at different timepoints within each experiment, statistical analysis of these data was performed using a linear mixed model to take into account this nestedness, including the experiment as a random effect, to take into account the correlation between the data. The statistical significance of differential expression between wild-type and mutant clones was assessed by fitting a linear mixed model to raw values of ΔCT with the genotype, the time and their interaction as fixed effects, and the experiment as a random effect. P values were corrected for multiple comparison using the Benjamini-Hochberg procedure (libraries: lme4, lmerTest, emmeans, corrplot; Data preparation in Supplementary Data 24–29, then Supplementary Data 18–23).

### Reporting summary

Further information on research design is available in the Nature Portfolio Reporting Summary linked to this article.

## Data availability

The datasets generated in this study are described in the Supplementary Information/Source Data files and have been deposited in the https://recherche.data.gouv.fr database under doi code [https://doi.org/10.57745/HSOOVU]. The raw human genomic data were protected and are not available due to data privacy laws. The genomic variant data used in this study are available in the ClinVar database under accession codes SCV004697335 and SCV004697336 [https://www.ncbi.nlm.nih.gov/clinvar/]. Source data are provided with this paper.

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

## Acknowledgements

We thank Dr. Mark Jobling and Dr. Yahya Khubrani for helping us with haplogroup identification. We thank Pascal Campagne from the Institut Pasteur Bioinformatics and Biostatistics Hub for his help with the statistical analysis. We thank Florence Deknuydt and the CYTO-ICAN platform from the ICAN Institute in Paris, France. We also thank Dr. Deepak Modi (NIRRH, India), Dr. Firuza Parikh (JHRC, India), and the members of the families who voluntarily donated the blood sample for our research work. This research was supported in part by grants from Agence Nationale de la Recherche (ANR-10-LABX-73 REVIVE, ANR-19-CE14-0022, ANR-19-CE14-0012, ANR-20-CE14-0007, ANR-23-CE14-0061, and ANR-23-CE14-0068) to A.B. and K.M.; Supported (in part) by ESPE Visiting Professorship, an unrestricted grant from Pfizer to K.M.; Powered@NLHPC: the supercomputing infrastructure of the NLHPC (CCSS210001), and ANID-BASAL FB210008 to C.F.L.; Wellcome Trust (209328/Z/17/Z) and National Institute for Health Research Great Ormond Street Hospital Biomedical Research Centre (IS-BRC-1215-20012) to F.B. and J.C.A.; and programs "Investissements d'avenir" ANR-10- IAIHU-06 and ANR11-INBS-0011 – NeurATRIS: Translational research Infrastructure for Biotherapies in Neurosciences to S.B. and D.B. For the purpose of open-access, the author has applied a CC-BY public copyright license to any author-accepted manuscript (AAM) version arising from this submission (D.H., C.E., J.B.-T., M.E., A.B., K.M., and J.C.A.).

## Author contributions

Conceptualization: D.H., K.M., A.B., J.C.A.; Data curation and formal analysis: D.H.; Funding acquisition: K.M., A.B., C.F.L., D.B. and J.C.A.; Investigation: D.H., C.E., J.B.T., M.E., I.G., V.L., L.S., S.W., C.F.L., F.B. and S.B.; Methodology: D.H., C.E., S.B. and D.B.; Project administration: D.H.; Resources: P.M., J.K., A.A. and G.S.C.; Supervision: D.H., A.B. and K.M.; Validation: D.H.; Visualization: D.H. and C.F.L.; Writing —original draft: D.H. and K.M.; Writing—review and editing: J.C.A., D.B. and A.B.

## Competing interests

The authors declare no competing interests.

## Additional information

[1]Institut Pasteur, Université Paris Cité, Human Developmental Genetics Unit, F-75015 Paris, France. [2]Centre National de la Recherche Scientifique, CNRS, UMR 3738 Paris, France. [3]Chemical Biology & Drug Discovery Lab, Escuela de Química y Farmacia, Facultad de Medicina y Ciencia, Universidad San Sebastián, Campus Los Leones, Lota 2465 Providencia, 7510157 Santiago, Chile. [4]Centro Ciencia & Vida, Fundación Ciencia & Vida, Av. del Valle Norte 725, Huechuraba 8580702 Santiago, Chile. [5]Institut Pasteur, Université Paris Cité, Epigenomics, Proliferation, and the Identity of Cells Unit, F-75015 Paris, France. [6]Institut Pasteur, Université Paris Cité, Stem Cells and Development Unit, F-75015 Paris, France. [7]Institut Pasteur, Université Paris Cité, Bioinformatics and Biostatistics Hub, F-75015 Paris, France. [8]Department of Pediatrics, Faculty of Medicine Ramathibodi Hospital, Mahidol University, Bangkok, Thailand. [9]Department of Assisted Reproduction and Genetics, Jaslok Hospital and Research Centre, Mumbai, India. [10]Department of Pediatric Surgery, Jaslok Hospital and Research Centre, Mumbai, India. [11]Genetics and Genomic Medicine Research & Teaching Department, UCL GOS Institute of Child Health, University College London, London, United Kingdom. [12]ICV-iPS core facility, Sorbonne Université, Institut du Cerveau - Paris Brain Institute - ICM, Inserm, CNRS, APHP, Hôpital de la Pitié Salpêtrière, Paris, France. [13]Institute for Women's Health, University College London, London, United Kingdom. [14]Sorbonne Université, Institut du Cerveau - Paris Brain Institute - ICM, Inserm, CNRS, APHP, Hôpital de la Pitié Salpêtrière, Paris, France. [15]Present address: Institut Cochin, Université Paris Cité, INSERM, CNRS, Paris, France. ✉e-mail: denis.houzelstein@cnrs.fr; kenneth.mcelreavey@pasteur.fr

