## [Peer Review File · Nature Communications]

A conserved NR5A1-responsive enhancer regulates SRY in testis-determinationReviewer #1 (Remarks to the Author):

In this manuscript entitled "A conserved NR5A1-responsive enhancer regulates SRY in testis-determination", Denis Houzelstein et al. discovered an evolutionary conserved SF-1/NR5A1-binding motif within a 250 bp region located 5 kb upstream of the SRY transcription start site. Furthermore, the authors identified unique single-base substitutions within the SF-1/NR5A1-binding element in 46,XY DSD patients. Although the findings of the manuscript are intriguing and would shed light on understanding the molecular pathology of 46XY,DSD, the manuscript has some flaws that prevent publication in the journal.

Major concerns,

#1

Figure 2A:

The DNAase seq data of Fig. 2A was obtained from embryonic human testis from a pool of healthy male fetuses. However, the stage of the sample was not specified. Was the stage of the pooled sample consistent? If so, At which stage were the fetus samples collected? Generally, in human fetuses, SRY starts to express around 6 weeks of GA, and this is believed to be the critical stage for sex determination.

Figure 2C:

The meaning of the figure is not clear. In which species, the sequence conservation is maintained? What do 0 and 100% of the value mean?

#2

Figure 4E

2-1:

HEK293T cells which are derived from human fetal kidneys were used for the luciferase assay. The cells would not have the phenotypes of gonads. Did the author try the reporter assay using other types of cells?

2-2:

The figure is busy with too many asterisks. Just showing the asterisks between reference and variant1/variant2/ Δ NR5A1 in the +NR5A1 group would be sufficient to show that the element is essential for the enhancer activity.

#3

Figure 5

In this figure, the authors tried to show that the enhancer candidate E250 would be essential for the adequate expression of SRY in vivo, and its impaired function would result in 46,XY DSD.

3-1:

The expression level of NR5A1 should be measured and included in Figure 5A and 5B. Without showing that NR5A1 expression levels are identical between WT and Mutant, it is not possible to conclude that E250- Δ 33 is responsible for the reduced expression level of SRY and SOX9.

3-2

Why the authors did not perform ChIP analysis (or ChIP seq) for NR5A1 using their iPS cells? The experiment will strongly support the authors' speculation that NR5A1 up-regulates SRY expression through binding to E250

3-3

I cannot catch the "time" scale of Fig.5A.

Does the time (in days) represent the maturation from pluripotent iPS cells to matured Sertoli cells?

If so, the authors are encouraged to show the data of the cell phenotype at each stage, including the expression levels of the testicular and ovarian marker genes, e.g., FGF9, AMH, WNT4, RSPO1, and FOXL2.

3-4

Why did the author not make iPSC cells with a single amino acid substitution identified in the patients?

As the author discussed L353-357, testicular determination could be exceptional, because a single base substitution in the enhancer element would cause 46,XDSD. If it is true, showing the single base substitution deteriorates the SRY expression in the iPSC model would be essential.

3-5

Fig 5B: The right graph that is carried out at 3.5 days, seems to be based on triplicate data (lacking the data of the rhombus dots), while the other two graphs have four times repeated data. The authors should explain the reason.

3-6

This reviewer is wondering if the story of the manuscript is true, the single base substitution would change the chromosomal 3D structure, and analyses of 3-D chromosome and genome structures, such as HiC, using induced Sertoli cells with the E250 deletion would provide valuable insight.

Minor concerns

L68-69: Differences/Disorders of Sex Development (DSD)

>> 46,XY Differences/Disorders of Sex Development (DSD), would be better

L137-L165: Evolution story can be moved to the Discussion section. The paragraph is too long and wordy.

Some "Nr5a1" should be written in NR5A1, e.g., L176, L178, L179.

L199: "25" was written in superscript. "Ref. 25" would be better?

Reviewer #2 (Remarks to the Author):

In humans, gonadal sex determination is regulated by complex and tightly controlled pro-testicular and pro-ovarian programs, alterations or disruptions of which can result in disorders of sex development (DSD). Despite decades of research, the majority of DSD patients with gonadal abnormalities have yet to receive a clinical genetic diagnosis. Since both the timing and the threshold levels of gene expression are key for testis/ovary development, it has been hypothesized that pathogenic variants in regulatory elements of key sex determining genes may be an important contributor to the etiology of DSD.

Houzelstein et al describe a beautiful study in which they identified the first known enhancer of the mammalian testis-determining gene SRY located within a 301 region of open chromatin, ~5 kb upstream of SRY. It defines also a new cause of SRY-linked 46,XY DSD, where a single-base pair substitutions in this 250 bp enhancer element of SRY result in a spectrum of 46,XY DSD phenotypes. This work is based on a combination of approaches and robust data such as computational analysis, modelling, experimental data and human genetics.

Here are my comments:

- 1) What about the other elements that surround the E250 motif. TFs do not act alone and NR5A1 will certainly act in concert with other factors. Could you describe in more detail the E250 and all known TF binding sites potentially affecting SRY expression.
- 2) Sertoli-like cell differentiation from hiPSCs (Figure 5). They developed in house a robust model of in vitro sequential differentiation of hiPSCs toward Sertoli. It would be important to evaluate the expression profile of SRY as well as SOX9 at all 7 stages instead of only three. In addition, it would be relevant to assess AMH expression as well to evaluate Sertoli cell differentiation status.
- 3) Phenotypic variability: the penetrance of the pathogenic variant is only 37%. If the DSD phenotype is indeed caused by a reduction of SRY expression due to the pathogenic variant in the NR5A1 binding site of the SRY E250 enhancer, then down-regulation of SRY should not be observed in hiPSCs from patients without DSD phenotype. Specifically, can the expression of SRY, SOX9 and AMH be tested with hiPSCs from patients with variant 1 or 2 exhibiting or not a DSD

phenotype?

Minor comments

- 1) Line 64: Sox9 and Fgf9: if these are proteins, mouse or human, they should be in capital letters. This remark is valid for Nr5a1 (lines 176-179)
- 2) Fig 1: include the TSS for SRY and exons
- 3) Fig 2: in the legend E800 and E250 are mentioned, but these regions are not clearly indicated in Fig 2A
- 4) Fig 2: Could you clarify in the legend of Fig 2B why some NR5A1 binding sites are annotated with different size of red triangles (e.g. springhare or Sloth)?
- 5) Line 123: Regarding the Dnase-seq analysis, could you mention which developmental stages of human fetal testes were used. Figure 3A: legend for 46, XY individuals raised as female (circle) or male (square) is not clearly visible. Please modify.

Reviewer #3 (Remarks to the Author):

The authors report a multi-dimensional study describing the SRY region and promoters that maintain evolutionary conservation. They report that NR5A1 serves the role as an enhancer in SRY testis determination. Specifically, the report mutations identified in a cohort of patients with gonadal dysgenesis that correspond to the NR5A1 gene. The reduced binding fitness is demonstrated first in silico then validated in vitro with corresponding mutations and a luciferase assay to report down regulation of SRY and SOX9 expression relative to wild type hiPSCs undergoing differentiation into Sertoli cells.

In the introduction, it is discussed that "Sertoli cells orchestrate the programs of cell-cell communication, migration, and differentiation leading to testis differentiation". Please discuss in the context of recent human single cell sequencing studies suggesting fate differentiation from interstitial progenitors prior to SRY expression in the sertoli lineage. (See Guo et al. 2020)

-Figure 5 - it would be helpful to designate the shapes of the experiments and which mutation they corresponded to.

-How do the authors explain the lack of significant difference at time points M2 and M3 for SRY? Presumably, lack of significance for SOX9 at M1 is due to SOX9 being a downstream target of SRY - can the authors comment and elaborate?

-it could be beneficial to also demonstrate the relative changes in WNT4 expression in the mutated and wild-type hiPSC model, since SOX9 is proposed to suppress pro-ovarian Wnt4 expression.

For the hiPSCs - line 724 states that 1.8% of the clones had the 33bp deletion - were these further selected for regarding downstream assays?

If so how?

If not, it would be difficult to interpret results if 95% of cells have off-target mutations from the CRISPR modification and only 1.8% containing the on-target mutation. Please elaborate and explain further.

Overall, this is a valuable series of experiments delineating testis development and mechanisms of dysfunction leading to gonadal dysgenesis.

We thank the reviewers and the editor for their constructive comments on the manuscript. Before detailing the reply to each comment, it is important to emphasize the unique aspects of mammalian sex-determination that have a direct impact on the interpretation of our data.

A wealth of data indicates that the initiation of testis-determination shows a remarkable lack of robustness.

Cell proliferation inhibitors in mouse organ culture models indicates that there is an 8-h window (10.8-11.2 dpc) that overlaps with a transient *Sry* gene expression, which is required for establishing testis-determination (PMID 12798287). An *Sry* transgene model has further refined this critical window to a 6-h interval (11.0 -11.25 dpc; PMID 19036799). A delay in reaching peak expression levels (PMIDs 35315790, 23102580, 27009039, 15680364), a delay in the initiation of expression (after 11.3 dpc; PMID 19036799) or premature upregulation of pro-ovarian genes before *Sry* expression results in XY sex-reversal (PMID 37917714). A twofold or less reduction in murine *Sry* expression is sufficient to cause XY sex-reversal (12750339).

Studies of human XY testicular dysgenesis also show that gene expression threshold levels are critical for correct testis-determination (PMID 35249806, 35957822). The study of cryptic changes in the SRY protein, observed in rare familial cases of human sex-reversal that is transmitted by fertile fathers, have demonstrated that SRY function is exquisitely sensitive to critical threshold levels (24003159, 35957822, 36568077). The commitment of the gonadal supporting cell lineage to Sertoli cells is also highly dependent on a threshold density of pre-Sertoli cells (3268409, 36912416). Sertoli cell induction is initiated by the expression of *SRY* in a small number of cells (<250) in the center of the gonad (PMIDs 11376487, 15680364, 36912416). *SRY* directly activates *SOX9* (PMIDs 15385158, 32886743) as well as paracrine factors such as FGF9 and prostaglandins (PMIDs 16185683, 17277314, 20040496, 25269616, 32886743). They, in turn, can induce Sertoli cell fate in neighboring cells in a density-dependent manner by upregulating *Sox9* expression independently of *Sry* (PMIDs 1769333, 25269616, 36912416). These data highlight that ***the initiation of testis-determination by SRY occurs in a small number of supporting cells and requires a minimum threshold level of SRY expression to be reached within a critical, restricted developmental window.***

The study of early human gonad formation and the functional analysis of variants causing human sex-reversal was hampered by the ***lack of a suitable animal or cellular model.*** The *in vitro* cellular reprogramming model that we developed was recently published (Science Advances 2023 PMID 36598988) and represents a major breakthrough in the study of mammalian sex-determination. The manuscript has been downloaded over 6.3K times from the Science website and reported by 6 news agencies, and cited by 4 publications.

Specific replies to Reviewers –

Reviewer #1

Major concerns,

Figure 2A: The DNase seq data of Fig. 2A was obtained from embryonic human testis from a pool of healthy male fetuses. However, the stage of the sample was not specified. Was the stage of the pooled sample consistent? If so, at which stage were the fetus samples collected? Generally, in human fetuses, *SRY* starts to express around 6 weeks of GA, and this is believed to be the critical stage for sex determination.

RI-1. We contacted Wouter Meuleman, first author of an article also describing the DNase seq data (PMID 32728217). He kindly shared the information that, unlike in the fetal testis where E250 is clearly accessible, accessibility was generally very modest at best, with weak detectability demonstrated in only two out of the 733 biological samples they studied. It is therefore tempting to propose that the lack of accessibility of E250 in most of the tissues is required for the precise expression of *SRY* in the gonad. We also contacted Luz Garcia-Alonso and Roser Vento-Tormo, authors of the article describing ATAC-seq data from single cells at the moment of human

sex-determination (PMID 35794482). They confirmed to us that E250 is accessible at 6-7 weeks post-conception which is precisely when *SRY* is expressed. The text has been altered accordingly.

Figure 2C:

The meaning of the figure is not clear. In which species, the sequence conservation is maintained? What do 0 and 100% of the value mean?

RI-2. Where sequences could be aligned, a 100% conservation means that a nucleotide is conserved in all the 18 sequences. A conservation of 50% would mean that for a given aligned nucleotide, it is conserved in only half of the species. The figure legend has been amended to clarify this point.

Figure 4E HEK293T cells which are derived from human fetal kidneys were used for the luciferase assay. The cells would not have the phenotypes of gonads. Did the author try the reporter assay using other types of cells?

RI-3 The luciferase assay has been successfully replicated in HeLa cells, yielding results that closely align with those obtained in HEK293T cells (see Supplementary files S15).

The figure is busy with too many asterisks. Just showing the asterisks between reference and variant1/variant2/ Δ NR5A1 in the +NR5A1 group would be sufficient to show that the element is essential for the enhancer activity.

RI-4. The figure has been modified accordingly.

Figure 5. In this figure, the authors tried to show that the enhancer candidate E250 would be essential for the adequate expression of *SRY* in vivo, and its impaired function would result in 46,XY DSD. The expression level of NR5A1 should be measured and included in Figure 5A and 5B. Without showing that NR5A1 expression levels are identical between WT and Mutant, it is not possible to conclude that E250- Δ 33 is responsible for the reduced expression level of *SRY* and *SOX9*.

RI-5. We have performed these experiments in both WT and mutant cells. We demonstrate that *NR5A1* is expressed during the appropriate stages of differentiation, and no differences were observed between the wild type and mutant cell lines. The Quantitative PCR results are provided in the supplementary files Fig05_Data_S16 to Fig05_Data_S22. These results are summarized in Fig05_Data_S22_01_PRISM_Compilation_ddCT.pzfx. Additionally, a dedicated pdf file presenting expression of all the markers, including *NR5A1*, is included as Fig05_Data_S22_02_Fig05.pdf. In the original submission we observed a modest but nonetheless significant decrease in *SOX9* expression in the mutant cell line compared to the wild-type cells. Additional biological replicates have mitigated this effect. However, with the additional replicates we continue to observe a robust and consistent reduction in the expression of *SRY* between the mutant and wild-type lines.

Why the authors did not perform ChIP analysis (or ChIP seq) for NR5A1 using their iPS cells? The experiment will strongly support the authors' speculation that NR5A1 up-regulates *SRY* expression through binding to E250

RI-6. This question has been addressed in point E-2 to the editor above. *In vivo*, single cell sequencing data confirms previous reports (PMIDs 11376487, 15680364, 36912416), which indicate that the *SRY* gene is expressed in a limited number of cells in the developing human (PMID 35794482) or mouse gonad (PMID 30893600). In our cellular hiPSC differentiation model, the expression levels of both *NR5A1* and *SRY* are low, approximately 30 C_T for *SRY* and about 32-33 C_T for *NR5A1* with the lower limit of detection by qPCR considered to be 35 C_T. In a Western blot analysis using differentiated hiPSCs, the expression level of *NR5A1* was also at the lower limit of detection. This result is consistent with the current model of testis-determination, where *SRY* is transiently expressed in a very limited number of cells and Sertoli cell differentiation occurs at least in part through the

paracrine recruitment of neighboring cells that do not express *SRY* (PMIDs 16185683, 17277314, 20040496, 25269616, 32886743). We repeatedly performed ChIP-PCR on differentiating cells in our model system but the data was uninformative (Annex 1). The most probable explanation for this is the small number of cells where NR5A1 is inducing *SRY* expression that is required for Sertoli cell specification and subsequent testis-determination. The analysis of human single cell sequencing data (Data S11) from developing human fetal gonads shows a low number of cells expressing *SRY* (<https://www.reproductivecellatlas.org/gonads/human-somatic/>). This data also shows the co-expression of *SRY* with *NR5A1*.

I cannot catch the “time” scale of Fig.5A. Does the time (in days) represent the maturation from pluripotent iPS cells to matured Sertoli cells?

RI-7. The timeline has been clarified in the figure.

If so, the authors are encouraged to show the data of the cell phenotype at each stage, including the expression levels of the testicular and ovarian marker genes, e.g., FGF9, AMH, WNT4, RSPO1, and FOXL2.

RI-8. We now provide a comprehensive description of the key factors involved in mammalian sex-determination (*SRY*, *SOX9*, *NR5A1*, *FGF9*, *WNT4*, *FOXL2*, *AMH*). These are described in Figure 5 and supplementary files S16 to S22. RSPO1 has not been included as it is not expressed in 46,XY cell lines in our model system (PMID 36598988).

Why did the author not make iPS cells with a single amino acid substitution identified in the patients?

RI-9. Two independent mutations were identified in these individuals. The first mutation (Variant-1) results in a Disorder of Sex Development (DSD) with variable expressivity and incomplete penetrance (Figure 3C; only 25% of carriers were raised as females). The second mutation (Variant-2) is predicted to be more severe and leads to a single case of complete sex reversal. However, we lack the father's DNA for study, and thus, we cannot determine whether the fertile father carries this variant.

The data from the large extended familial case of Y-linked sex-reversal strongly resembles the data of the classic B6.Y^{POS} mouse model of XY sex-reversal, where testis-determination defects in B6.Y^{POS} are associated with a subtle dysregulation of *Sry* expression (PMIDs 7089579, 15680364, 31836612). The degree of sex-reversal is variable and a proportion of mice are male and fertile depending on the genetic background (PMID 31836612). Protection against B6.Y^{POS} sex-reversal is mediated, at least in part, by enhanced *Sry* expression due to an autosomal locus on Chr13 (PMID 31836612). The human sex-reversed pedigree suggests, that the Y-linked sex-reversing variant alters the expression level of the *SRY* around a critical threshold. As a result of these observations, we decided to delete the NR5A1-binding site rather than introduce specific point mutations with potential residual activity. This decision was made to ensure a more robust and reliable readout in both *in vitro* and *ex vivo* assays. The deletion of the putative enhancer element shows a consistent and robust reduction in the expression *SRY* expression in the *in vitro* model.

As the author discussed L353-357, testicular determination could be exceptional, because a single base substitution in the enhancer element would cause 46,DSD. If it is true, showing the single base substitution deteriorates the *SRY* expression in the iPS model would be essential.

RI-10. This question is addressed in R1.9 above. We believe that the genetic evidence we provide is very robust, with two independent single base pair substitutions associated with sex-reversal in completely different genetic backgrounds, one of these substitutions segregating in a very large family. In this article, we also use a combination of *in silico* computational and *in vitro* cell-based analyses to evaluate the impact of these two variants on the biological activity of the E250 response element, which demonstrated the disruptive effects of these variants. As stated above we wished to demonstrate the importance of the NR5A1-binding element in the control of *SRY* expression and we decided to delete the entire element rather than introduce single base-pair changes.

Fig 5B: The right graph that is carried out at 3.5 days, seems to be based on triplicate data (lacking the data of the rhombus dots), while the other two graphs have four times repeated data. The authors should explain the reason.

RI-11. The differentiation experiment was repeated 5 times. Each experiment required processing of a substantial number of plates. Handling too many plates in parallel would have introduced heterogeneity between the first and last plates. For this reason, all timepoints were not included in every experiment. The reference timepoint (M1-36h) was present in every experiment, allowing for the comparison between different experiments. The statistical test employed (Linear Mixed Model) took into consideration the heterogeneity between the experiments. The detail of the experiments is given in supplementary files S16 to S22.

This reviewer is wondering if the story of the manuscript is true, the single base substitution would change the chromosomal 3D structure, and analyses of 3-D chromosome and genome structures, such as HiC, using induced Sertoli cells with the E250 deletion would provide valuable insight.

RI-12. The single base substitutions are therefore predicted to specifically impact the binding of the NR5A1 transcription factor to the E250 enhancer leading to a reduced expression of the *SRY* gene. This could lead to changes in the 3D structure resulting in altered expression. This aspect was not the aim of the current study but is of interest for future studies on the entire *SRY* locus. In this study we have demonstrated that an NR5A1 binding element is key for *SRY* expression, however other elements including 3' elements are likely to be involved (e.g. we previously published a deletion 3' to *SRY* causing XY sex-reversal PMID 8710915). These conformational studies would be of interest in a finer and comprehensive analysis of the entire *SRY* locus during testis differentiation.

Minor concerns

L68-69: Differences/Disorders of Sex Development (DSD)- 46,XY Differences/Disorders of Sex Development (DSD), would be better

RI-13. The text has been amended.

L137-L165: Evolution story can be moved to the Discussion section. The paragraph is too long and wordy.

RI-14. We understand the concerns of the reviewer. However, we consider that the evolutionary aspects (L137-L165) of the *SRY* locus is an integral component of the study. The evolutionary aspects of this work, placed in the results section, plays a crucial role in providing the necessary context for the data presented in all the subsequent sections including why we choose our *in vitro* cellular reprogramming model and why we cannot complement our data with murine studies. We are concerned that moving these aspects to the discussion section will disrupt the overall coherence of the study.

Some “Nr5a1” should be written in NR5A1, e.g., L176, L178, L179.

RI-16. The text has been corrected.

Reviewer #2:

What about the other elements that surround the E250 motif. TFs do not act alone and NR5A1 will certainly act in concert with other factors. Could you describe in more detail the E250 and all known TF binding sites potentially affecting SRY expression.

R2-1. The exhaustive list of transcription factors predicted to bind E250 is provided in the supplementary file "Fig02_Data_S06_250_predicted_transcription_binding_sites_Genomatix_250.xlsx." The binding sites of transcription factors, including SOX9 and NR5A1 (SF1), are highlighted in red.

Sertoli-like cell differentiation from hiPSCs (Figure 5). They developed in house a robust model of in vitro sequential differentiation of hiPSCs toward Sertoli. It would be important to evaluate the expression profile of SRY as well as SOX9 at all 7 stages instead of only three. In addition, it would be relevant to assess AMH expression as well to evaluate Sertoli cell differentiation status.

R2-2. Figure 5 has been completed with more data sets. The complete expression profile of *SRY*, *SOX9*, *WNT4*, *FOXL2*, *WNT4*, *NR5A1*, *FGF9* and *AMH* is given as supplementary material (Fig05_Data_S16 to Fig05_Data_S22). In the main figure 5 we have focused on the expression of *SRY*, *SOX9* and the pro-ovarian marker *WNT4* for clarity. The key element here is the initiation of the expression of *SRY* where a consistent and significant reduction in the expression of *SRY* is observed.

Phenotypic variability: the penetrance of the pathogenic variant is only 37%. If the DSD phenotype is indeed caused by a reduction of SRY expression due to the pathogenic variant in the NR5A1 binding site of the SRY E250 enhancer, then down-regulation of SRY should not be observed in hiPSCs from patients without DSD phenotype. Specifically, can the expression of SRY, SOX9 and AMH be tested with hiPSCs from patients with variant-1 or -2 exhibiting or not a DSD phenotype?

R2-3. This is an interesting point. We successfully generated hiPSCs, where the NR5A1-binding motif is deleted. A comparison was performed between this modified cell line and the wild-type cell line from which it was derived. The reason for this is simple. We do not have access to biological material from different family members in the two extended pedigrees. The individual carrying Variant-2 is no longer available for further studies. Whilst we agree that testing hiPSCs from the affected patients could be interesting, these bottlenecks when dealing with human material are precisely why we developed an *in vitro* model, where we could test different pathological variants using the same cell line. The question also alludes to the possibility that the variability in the phenotype in the familial case may be due to genetic modifiers in individuals in the family that do not show sex-reversal. As mentioned in the reply R1-9 above the murine B6.Y^{POS} sex-reversal model is a good example where disrupted *Sry* expression can be compensated by an independent autosomal locus (a good candidate is *Gadd45g*, PMID 35315790). This may be the situation in the familial case of sex-reversal but we do not have the material to test this. Conversely, the XY sex-reversal may simply be a threshold effect of *SRY* expression without the need to invoke other loci. A twofold or less reduction in murine *Sry* expression is sufficient to cause sex-reversal (12750339). Rare familial cases of human sex-reversal caused by missense variants in the SRY protein and transmitted by fertile fathers have been described, where the biological activity of the protein is at the balance of a critical threshold level (24003159, 35957822, 36568077). Sex-reversal in the Y-linked familial case in this study may well be due to stochastic variability in expression levels that may not be reproduced using hiPSCs derived from unaffected individuals.

Minor comments

Line 64: Sox9 and Fgf9: if these are proteins, mouse or human, they should be in capital letters. This remark is valid for Nr5a1 (lines 176-179)

R2-4. Thank you for bringing up this point, which has also been raised by another reviewer. The term "Nr5a1" had been employed when referring to the mammalian protein in a general context, encompassing not only the human or murine versions. We have replaced all instances of "Nr5a1" with "NR5A1" throughout the document. This change ensures consistency and clarity in our references to the protein.

Fig 1: include the TSS for SRY and exons

R2-5. The transcription start site (TSS) for *SRY* has been defined. It is important to note that the *SRY* gene in humans consists of a single exon. The entire annotated sequence of the *SRY* gene locus can be found in the supplementary files: Fig01_Data_S01_supporting_data_for_Fig1A.clc and Fig01_Data_S02_supporting_data_for_Fig1B.clc.

Fig 2: in the legend E800 and E250 are mentioned, but these regions are not clearly indicated in Fig 2A

R2-6. The figure has been modified accordingly.

Fig 2: Could you clarify in the legend of Fig 2B why some NR5A1 binding sites are annotated with different size of red triangles (e.g. springhare or Sloth)?

R2-7. This was an error in the triangle size in this figure. The size of each triangle is now identical.

Line 123: Regarding the Dnase-seq analysis, could you mention which developmental stages of human fetal testes were used.

R2-8. This has also been addressed in point R1-1 to reviewer 1. We contacted Wouter Meuleman, first author of an article also describing the DNase seq data (PMID 32728217). He kindly shared the information that, unlike in the fetal testis where E250 is clearly accessible, accessibility was generally very modest at best, with weak detectability demonstrated in only two out of the 733 biological samples they studied. It is therefore tempting to propose that the lack of accessibility of E250 in most of the tissues is required for the precise *SRY* in the gonad. We also contacted Luz Garcia-Alonso and Roser Vento-Tormio, authors of the article describing ATAC-seq data from single cells at the moment of human sex-determination (PMID 35794482). They confirmed us that E250 is accessible at 6-7 weeks post-conception which is precisely when *SRY* is expressed. The text has been altered accordingly.

Figure 3A: legend for 46, XY individuals raised as female (circle) or male (square) is not clearly visible. Please modify.

R2-8. The color of the circle has been changed to increase the contrast.

Reviewer #3:

In the introduction, it is discussed that "Sertoli cells orchestrate the programs of cell-cell communication, migration, and differentiation leading to testis differentiation". Please discuss in the context of recent human single cell sequencing studies suggesting fate differentiation from interstitial progenitors prior to *SRY* expression in the sertoli lineage. (See Guo et al. 2020)

R3-1. A reference to Guo et al. 2021 has been introduced and the text amended accordingly in the introduction "Single cell sequencing analysis of human developing gonadal cells indicate that Sertoli and interstitial cells originate from a common heterogeneous progenitor pool, which then resolves into fetal Sertoli cells or interstitial cells that include Leydig cells. The data suggests that in the human Leydig and Sertoli cell specification occurs at or near the same developmental time."

Figure 5 - it would be helpful to designate the shapes of the experiments and which mutation they corresponded to.

R3-2. The figure has been modified. The experimental details, including qPCR raw results, $\Delta\Delta\text{CT}$ analysis, outlier identification, and statistical tests, along with a summary of the results and graphical representation, are provided in Fig05_Data_S16 to Fig05_Data_S22.

How do the authors explain the lack of significant difference at time points M2 and M3 for *SRY*? Presumably, lack of significance for *SOX9* at M1 is due to *SOX9* being a downstream target of *SRY* - can the authors comment and elaborate?

R3-3. The cellular model of gonadal-like cell differentiation that we developed has been comprehensively described (PMID 36598988). Throughout the differentiation process, the cells display sustained expression of testis-specific genes, undergo migration, and form tubular structures. We now show using this model that in the wild-type cells we observe a transient increase in *SRY* expression followed by an increase in *SOX9* expression, as expected for an *SRY* direct target. We postulate that the E250 NR5A1-binding site plays a critical role in initiating the transient increase in *SRY* expression. A delay in reaching peak *SRY* expression levels is analogous to the situation described in the B6.Y^{POS} mouse model of XY sex-reversal, where testis-determination defects in B6.Y^{POS} are associated with a subtle delay in *Sry* expression (PMIDs 7089579, 15680364, 31836612). This is associated with a delay in *Sox9* expression (31836612). The degree of sex-reversal is variable and a proportion of mice are male and fertile depending on the genetic background (31836612). *Sry* expression must reach a threshold level in cells of the supporting cell lineage within a certain developmental window of time to initiate testis determination. If this threshold is exceeded too late then proper testis-determination does not occur. The observations in the B6.Y^{POS} mouse model closely resemble our findings with modelling the deletion of the NR5A1 enhancer element and the range of phenotypes seen in the familial case of sex-reversal.

It could be beneficial to also demonstrate the relative changes in WNT4 expression in the mutated and wild-type hiPSC model, since SOX9 is proposed to suppress pro-ovarian Wnt4 expression.

R3-4. This was performed. Please refer to answer to reviewer R1-8. The expression of *WNT4*, *FOXL2*, *NR5A1*, *AMH* has also been quantified. These are described in Figure 5 and supplementary files S16 to S22. Apart from the significant reduction in *SRY* expression at M1-36, there was no other significant changes in the expression of *SOX9*, *WNT4*, *FOXL2*, *NR5A1* nor *AMH*.

For the hiPSCs - line 724 states that 1.8% of the clones had the 33bp deletion - were these further selected for regarding downstream assays? If so how?

R3-5. The platform that generated the CRISPR-CAS9 edited cell lines provided two modified clones that were systematically validated for the absence of genome anomalies. Both of these clones yielded qualitatively similar results in our assays. For the sake of homogeneity, we are presenting results obtained with one of them. Every cell in each of the clone line carries the same targeted mutation, there is therefore no genomic heterogeneity in the cells as determined by iCS-digitalTM PSC – 24 probes to check the most common genomic abnormalities before and after the modification (StemGenomics, Montpellier, France).

If not, it would be difficult to interpret results if 95% of cells have off-target mutations from the CRISPR modification and only 1.8% containing the on-target mutation. Please elaborate and explain further.

R3-6 As answered in R3-5, every cell in each of the clone lines carries the same targeted mutation, there is therefore no heterogeneity in the population on which the experiments were performed.

Annex 1 Summary of ChIP-PCR data

Methodology and Results

Western Blot

The efficiency and specificity of the anti-NR5A1 antibody (#07-618; Merck) used for immunoprecipitation for CHIP-PCR was verified by performing western blot on Ntera2 (NT2D1) and HEK293T (HEK) cells transfected with a MYC-tagged NR5A1 expressing vector (#RG207577; Origene). This antibody is widely used for expression studies, co-IP and ChIP-seq analysis (PMIDs

21163858, 21087664, 22927646, 24205079, 23907384). The proteins were migrated on a 4–15% Criterion™ TGX Stain-Free™ Protein Gel, (#5678084; BioRad). (A) Three lanes were run in parallel with increasing quantities of the protein (10 µg, ≈20 µg and 30 µg). On performing the western blot using established protocols (PMID 24549039) we observed a band of ≈60kDa in all the transfected samples. The band was absent from the non-transfected cells. The observed band was slightly larger than the endogenous NR5A1 (≈52kDa) due to the presence of the Myc-tag. (B) To verify the presence of protein in all the lanes the blots were stripped of anti-NR5A1 antibody using Restore™ Western Blot Stripping Buffer (# 21059; ThermoFisher Scientific) using the manufacturer’s protocol. The blots were then re-probed with anti-BETA-ACTIN antibody (#A2228; Sigma). We observed a band of ≈42kDa in all transfected samples.

Cell lines

45 = Wild Type hiPS Cell line. 82 = Mutant hiPS Cell line derived from 45 in which a 33bp deletion has been introduced in the E250 enhancer (GRCh38:chrY:2,792,792-2,792,824).

Differentiation experiment

ChIP01_iPS23 — Cells harvested at M2+29h00 (*i.e.* 69 hours after the beginning of differentiation, a time when *SRY* expression was highest).

ChIP02_iPS24 — Cells harvested at M2+29h00 (*i.e.* 69 hours after the beginning of differentiation).

Taqman probes specific to the regions of interest :

Target	Reference	Target size (bp)	Amplicon (bp)	coordinates
SRY-E250	AREPUZ2	247	143	GRCh38:chrY:2792658-2792905
SRY-E250-2	ARAAEVT	179	125	GRCh38:chrY:2792609-2792788
WT1-2	AR7DTK7	149	146	GRCh38:chrY:2790772-2790921
Enh13	ARMF2D9	202	202	GRCh38:chr17:71484731-71484933

Amplicons:

1 – SRY-E250: This element, located on the Y chromosome, is hemizygous. The Taqman probe overlaps with the NR5A1-binding site. In the presence of the 33bp deletion of the enhancer element the probe cannot bind.

2 – SRY-E250-2: The Taqman probe is positioned immediately 5' to the SRY-E250 probe. The deletion does not affect the binding of the probe.

3 – WT1: The Taqman probe is positioned immediately 5' to the SRY-E250-2 probe, outside the enhancer of interest, its binding is not affected by the deletion. The amplified element does not contain an NR5A1-binding motif and should not be immunoprecipitated.

4 —Enh13: The Taqman probe is centered on a DNA fragment known to be a conserved enhancer of *SOX9* (located on an autosome), a target of *SRY*, and regulated by NR5A1, in both humans and mice (PMIDs 29903884, 30552336, 35921234). We observed that in our cellular model in WT cells, the expression of *SOX9* increases following the upregulation of *SRY* expression. NR5A1 binding to this site should not be affected by the mutation in E250.

ChIP-qPCR. Cells were crosslinked (50 min DSG at 2 mM; Sigma, 80424-5 mg, followed by 10 min with formaldehyde 1%; Thermo, 28908). After fixation, chromatin was prepared as previously described (Festuccia et al. 2016,. Mitotic binding of Esrrb marks key regulatory regions of the pluripotency network. Nat Cell Biol 18: 1139–1148. 10.1038/ncb3418) and sonicated with a Bioruptor Pico (Diagenode). Chromatin was pre-cleared and immunoprecipitated with anti-NR5A1 (Rabbit Anti-SF1 polyclonal antibody (Sigma Aldrich, 07-618) antibody overnight rotating on-wheel at 4 °C in 500 µl of TSE150. Twenty microlitres was set apart for input DNA extraction and precipitation. Twenty-five microlitres of blocked pG beads 50% slurry was added for 4 h rotating on-wheel at 4 °C. Beads were pelleted and washed for 5 min rotating on-wheel at room temperature with 1 ml of buffer in the following order: 3 × TSE150, 1 × TSE500 (as TSE150 but 500 mM NaCl), 1× washing buffer (10 mM Tris-HCl pH8, 0.25M LiCl, 0.5% NP-40, 0.5% Na-deoxycholate, 1 mM EDTA), and 2 × TE (10 mM Tris-HCl pH8, 1 mM EDTA). Elution was performed in 100 µl of elution buffer (1% SDS, 10 mM EDTA, 50 mM Tris-HCl pH 8) for 15 min at 65 °C after vigorous vortexing. Eluates were collected after centrifugation and beads rinsed in 150 µl of TE-SDS1%. After centrifugation, the supernatant was pooled with the corresponding first eluate. For both immunoprecipitated and input chromatin, the crosslinking was reversed overnight at 65 °C, followed by proteinase K treatment, phenol/chloroform extraction and ethanol precipitation.

ChIP01

		1 - Normalizer	2 – SRY-E250-2	3 – WT1-2 Negative control	4 – SRY-E250 WT specific
CT	IP45-2303	33,70	34,06	33,70	35,07
	IP82-2304	33,12	33,88	33,42	40,00
	Input45-2305	29,26	31,09	30,33	31,78
	Input82-2306	28,70	31,75	30,02	40,00
ΔCT	IP45-2303		0,35	0,00	1,37
	IP82-2304		0,76	0,30	6,88
	Input45-2305		1,83	1,07	2,51
	Input82-2306		3,05	1,32	11,30
ΔΔCT	IP45-2303		1,00	1,28	0,49

IP82-2304		0,75	1,04	0,01
Input45-2305		0,36	0,61	0,22
Input82-2306		0,15	0,51	0,00

- 1- The number of copies detected in the input DNA is in the same order of magnitude in (1), (2), and (3). It is slightly higher for the normalizer (1), likely due to its location on an autosome.
- 2- The genotype of the cell lines is confirmed by SRY-E250 (4).
- 3- There is no difference between the amount of precipitated DNA between the positive (1) and negative (2) control.
- 4- There is no major difference in the amount of precipitated DNA between ED250 (2) and the negative control (3).
- 5- There is no major difference in the amount of precipitated DNA between the wild type and mutant cell lines.

ChIP02

		1 - Normalizer	2 – SRY-E250-2	3 – WT1-2 Negative control	4 – SRY-E250 WT specific
CT	IP45-2303	34,69	34,41	34,29	-
	IP82-2304	33,88	33,92	33,59	-
	Input45-2305	29,76	30,76	30,80	31,94
	Input82-2306	27,98	27,91	27,81	36,91
Δ CT	IP45-2303		-0,28	-0,41	-
	IP82-2304		0,04	-0,29	-
	Input45-2305		1,01	1,04	2,18
	Input82-2306		-0,07	-0,17	8,94
$\Delta\Delta$ CT	IP45-2303		1,00	1,09	-
	IP82-2304		0,80	1,01	-
	Input45-2305		0,41	0,40	0,18
	Input82-2306		0,86	0,92	0,00

- 1- The number of copies detected in the input DNA is in the same order of magnitude in (1), (2), and (3). It may be slightly higher for the normalizer (1), likely due to its location on an autosome.
- 2- The genotype of the cell lines is confirmed by SRY-E250 (4).
- 3- There is no difference between the amount of precipitated DNA between the positive (1) and negative (2) control.
- 4- There is no major difference in the amount of precipitated DNA between ED250 (2) and the negative control (3).

- 5- There is no major difference in the amount of precipitated DNA between the wild type and mutant cell lines.

Reviewer #1 (Remarks to the Author):

The authors responded to each comment appropriately.

Reviewer #2 (Remarks to the Author):

Most of my concerns have been addressed and the revised version deserves to be published in Nat Comm.

Regarding other TFs that would act in concert with NR5A1 on the E250 enhancer, I still think it would be relevant to include in the discussion information about other potential TFs that bind and promote SRY expression (and not just in the supplementary figure 2).

Regarding my small comment on Figure 2A and the annotation of E800 and E250. Although the text mentioning them is in the figure, it's so small you can't read it. To be resized.

Reviewer #3 (Remarks to the Author):

The authors have appropriately addressed each reviewer's comments with supporting experimental data and literature references. I do not have any additional comments or suggested revisions.

We have replied to the minor comments of Reviewer 2

REVIEWERS' COMMENTS

Reviewer #1 (Remarks to the Author):

The authors responded to each comment appropriately.

Reviewer #2 (Remarks to the Author):

Most of my concerns have been addressed and the revised version deserves to be published in Nat Comm.

Regarding other TFs that would act in concert with NR5A1 on the E250 enhancer, I still think it would be relevant to include in the discussion information about other potential TFs that bind and promote SRY expression (and not just in the supplementary figure 2).

In the absence of comparative genomic analysis, human genetics data and functional studies, we do not wish to over speculate on the potential role of predicted TF binding sites within E250. We have modified the discussion slightly to include the phrase –

It is unlikely that NR5A1 acts alone to control SRY expression on the E250 enhancer. Other transcription factors may well act in concert with NR5A1 to promote SRY expression during testis-determination. The NR5A1-binding site is immediately flanked by two predicted consensus SOX-binding sites suggesting the possibility of a positive autoregulatory feedback loop enhancing SRY expression (Supplemental S2).

Regarding my small comment on Figure 2A and the annotation of E800 and E250. Although the text mentioning them is in the figure, it's so small you can't read it. To be resized.

This has been resized as requested.

Reviewer #3 (Remarks to the Author):

The authors have appropriately addressed each reviewer's comments with supporting experimental data and literature references. I do not have any additional comments or suggested revisions.